# Aplp1 interacts with Lag3 to facilitate transmission of pathologic α-synuclein

Xiaobo Mao [1,2,3,25] ✉, Hao Gu[1,2,16,17,25], Donghoon Kim [1,2,20,25], Yasuyoshi Kimura [1,2], Ning Wang [1,2], Enquan Xu [1,2], Ramhari Kumbhar[1,2,3], Xiaotian Ming[1,2], Haibo Wang [1,2], Chan Chen [1,2,18], Shengnan Zhang[4], Chunyu Jia[4,5], Yuqing Liu [1,2], Hetao Bian[1,2], Senthilkumar S. Karuppagounder[1,2], Fatih Akkentli[1,2,3], Qi Chen[1,2], Longgang Jia[1,2], Heehong Hwang[1,2], Su Hyun Lee[1,2], Xiyu Ke[6,7], Michael Chang [1,2], Amanda Li[1,2], Jun Yang [1,2], Cyrus Rastegar[1,2], Manjari Sriparna [1,2], Preston Ge [1,2,19,23,24], Saurav Brahmachari [1,2], Sangjune Kim [1,2,21], Shu Zhang [1,2], Yasushi Shimoda [8], Martina Saar[9], Haiqing Liu[1,2,22], Sin Ho Kweon[1,2], Mingyao Ying[2,10], Creg J. Workman [11], Dario A. A. Vignali [11,12], Ulrike C. Muller [9], Cong Liu [4], Han Seok Ko [1,2,3] ✉, Valina L. Dawson [1,2,3,13,14] ✉ & Ted M. Dawson [1,2,3,14,15] ✉

Pathologic α-synuclein (α-syn) spreads from cell-to-cell, in part, through binding to the lymphocyte-activation gene 3 (Lag3). Here we report that amyloid β precursor-like protein 1 (Aplp1) interacts with Lag3 that facilitates the binding, internalization, transmission, and toxicity of pathologic α-syn. Deletion of both Aplp1 and Lag3 eliminates the loss of dopaminergic neurons and the accompanying behavioral deficits induced by α-syn preformed fibrils (PFF). Anti-Lag3 prevents the internalization of α-syn PFF by disrupting the interaction of Aplp1 and Lag3, and blocks the neurodegeneration induced by α-syn PFF in vivo. The identification of Aplp1 and the interplay with Lag3 for α-syn PFF induced pathology deepens our insight about molecular mechanisms of cell-to-cell transmission of pathologic α-syn and provides additional targets for therapeutic strategies aimed at preventing neurodegeneration in Parkinson's disease and related α-synucleinopathies.

α-Synucleinopathies are a subset of neurodegenerative diseases[1], including Parkinson's disease (PD), dementia with Lewy Bodies (DLB), and multiple system atrophy (MSA), which are characterized by abnormal accumulation of misfolded α-synuclein (α-syn) in neurons or glial cells. Pathologic α-syn, which is a prion-like protein, progressively spreads from the enteric and peripheral nervous systems into the central nervous system[2]. Support for pathologic α-syn transmission is the appearance of Lewy body (LB) pathology in transplanted fetal mesencephalic dopaminergic neurons of PD patients[3–5].

In cell-to-cell transmission models of pathologic α-syn, exogenous recombinant α-syn preformed fibrils (PFF) enter neurons and seed endogenous α-syn monomers[6,7], leading to substantial α-syn pathology and neurotoxicity[8]. Intrastriatal or gut injection of α-syn PFF induces pathologic α-syn spreading, resulting in PD-like motor and non-motor symptoms[9–16].

The cell-to-cell transmission of pathologic α-syn through cell surface receptors and its subsequent uptake is considered a pivotal mechanism in the pathogenesis and dissemination of pathologic α-syn. These cell surface receptors are integral to the uptake and progression of pathological α-syn species across neural networks. Recent work has identified several receptors for pathologic α-syn uptake, including Lag3, and heparan sulfate proteoglycans (HSPGs) among others[17]. Lymphocyte-activation gene 3 (Lag3) is a receptor[18,19] that facilitates α-syn transmission from neuron to neuron[20] and plays a role

in α-syn PFF uptake in brain microvascular endothelial cells[21]. Depletion of Lag3 significantly reduces the internalization of α-syn PFF in neurons but fails to completely prevent the internalization of α-syn PFF, suggesting that other receptors and mechanisms for α-syn transmission likely exist[20,22–27]. Although amyloid β precursor-like protein 1 (Aplp1) is a pathologic α-syn binding protein[20] that belongs to the conserved amyloid precursor protein (App) family and is associated with neurodegenerative disorders[28–30], the role of Aplp1 in pathologic α-syn transmission and pathogenesis is not known.

In this study, we show that genetic deletion of Aplp1 significantly inhibits the internalization of pathologic α-syn, cell-to-cell transmission, neurotoxicity, and behavioral deficits induced by α-syn PFF. Furthermore, Aplp1 and Lag3 act as accessory proteins that drive pathologic α-syn transmission. Deletion of both Aplp1 and Lag3 eliminates the loss of dopaminergic (DA) neurons and the accompanying behavioral deficits induced by α-syn PFF. Disruption of the Aplp1-Lag3 interaction by using an anti-Lag3 antibody prevents pathologic α-syn transmission and pathogenesis providing a key route to prevent the degenerative process set in motion by pathologic α-syn. These findings indicate that both Aplp1 and Aplp1-Lag3 interaction play roles in the cell-to-cell transmission of pathologic α-syn and provide a proof of concept for Lag3-targeting immunotherapy. These observations may lead to the optimization of pathologic α-syn receptor-targeted therapies.

## Results

### α-Syn PFF binds to amyloid β precursor-like protein 1 (Aplp1)

We conjugated the α-syn monomer with biotin (α-syn-biotin)[20] and generated α-syn-biotin PFF using an established protocol[15]. We confirmed that α-syn-biotin PFF specifically binds to Aplp1-transfected SH-SY5Y cells in a saturable manner, with a $K_d$ of 430 nM (Supplementary Fig. 1a) by a well-established cell surface-binding assay as previously reported[20]. In contrast, the α-syn-biotin monomer at a concentration as high as 3000 nM does not exhibit any appreciable binding to Aplp1-transfected SH-SY5Y cells (Supplementary Fig. 1a). β-Amyloid-biotin PFF binds to SH-SY5Y cells expressing Aplp1 at high concentrations, and β-amyloid-biotin monomer does not exhibit any appreciable binding signal (Supplementary Fig. 1b). It is possible that β-amyloid-biotin PFF could bind to Aplp1 at higher concentrations and could saturate at higher concentrations. Moreover, α-syn-biotin PFF does not bind to the amyloid precursor protein (App) or the amyloid precursor-like protein 2 (Aplp2)[20], suggesting that the binding between Aplp1 and α-syn-biotin PFF is specific.

Like other App family proteins, Aplp1 is a single-pass transmembrane protein implicated in synaptic adhesion[31]. Aplp1 possesses a large N-terminal extracellular domain and a small intracellular C-terminal region[32]. To identify the α-syn PFF-binding domain, we sequentially deleted each of the four domains of Aplp1 (ΔE1, ΔAcD+E2, ΔJMR (juxtamembrane region)), and ΔICD (intracellular domain) (Fig. 1a). The membrane locations of different deletion mutants of FLAG-Aplp1 were determined first (Supplementary Fig. 1c, e) along with Lag3-Myc (Supplementary Fig. 1d, e) with the membrane marker wheat germ agglutinin (WGA) conjugated with Alexa Fluor™ 488. This was followed by determining the binding of α-syn-biotin PFF to the Aplp1 deletion mutants by a cell surface-binding assay (Fig. 1b, Supplementary Fig. 2a). These experiments suggest that: (i) α-syn-biotin PFF preferentially bind to the E1 domain, (ii) deletion of the AcD+E2 domain does not interfere with binding, (iii) deletion of JMR and the transmembrane (TM) domain substantially weakens the binding and serve as controls as these mutants lack cell membrane expression, and (iv) deletion of ICD mildly reduces the binding (Fig. 1b, Supplementary Fig. 2a). Since α-syn-biotin PFF binding to Aplp1 is eliminated by deletion of the E1 domain, we further examined the two subdomain deletion mutants of ectodomain E1, the growth factor-like domain (ΔGFLD) and the copper-binding domain (ΔCuBD). Deletion of CuBD

does not interfere with binding, while deletion of the GFLD subdomain moderately interferes with the binding (Fig. 1b, Supplementary Fig. 2a).

To confirm that the E1 domain of Aplp1 interacts with α-syn-biotin PFF, we constructed Aplp1 chimeras, in which E1-AcD of Aplp2 or App is replaced with the E1-AcD ectodomain of Aplp1 (Fig. 1c). Both Aplp1(E1)-Aplp2 and Aplp1(E1)-App chimeras exhibit binding to α-syn-biotin PFF, while Aplp2 and App do not bind to α-syn-biotin PFF (Fig. 1d, Supplementary Fig. 2b). Since the binding of α-syn-biotin PFF to the Aplp1(E1)-Aplp2 and Aplp1(E1)-App chimeras do not bind at the same level as Aplp1 (Fig. 1d, Supplementary Fig. 2b), the ICD subdomain of Aplp1 may help facilitate the interaction, which is consistent with the effect of deletion of ICD subdomain of Aplp1 on α-syn-biotin PFF binding to Aplp1.

We identified a seven-amino acid (7aa) motif with a similar sequence in Aplp1 and Lag3: GGTRSGR (121-127) in Aplp1 and the GGLRSGR (103–109) in Lag3. Interestingly this 7aa motif is not in App and Aplp2, which might account for their lack of binding to pathologic α-syn. The 7aa motif is located in the α-syn-biotin PFF-binding domains of Aplp1 (GFLD subdomain) (Fig. 1a) and Lag3 (D1 domain) (Fig. 1e). Deletion of this 7aa (Δ7aa) or the substitution with seven alanine (7aa-A × 7) of Aplp1 or Lag3 significantly reduced the binding to both receptors (Fig. 1b, f, Supplementary Fig. 2c). Taken together, these results suggest that the E1 domain of Aplp1 is the major binding domain for α-syn-biotin PFF, particularly depletion of GFLD subdomain and the 7aa motif in the E1 domain can significantly reduce α-syn-biotin PFF binding to Aplp1.

### Aplp1 mediates the endocytosis of α-syn PFF

To determine whether Aplp1 is involved in the endocytosis of α-syn PFF, we conjugated α-syn PFF to a pH-sensitive pHrodo red dye (α-syn-pHrodo PFF). As the pHrodo dye is non-fluorescent at neutral pH and fluoresces brightly in acidic environments, we used it to monitor the internalization of α-syn PFF as previously described[20]. We treated WT or Aplp1 knockout ($Aplp1^{-/-}$) primary cortical neurons at 12–14 days in vitro with α-syn-pHrodo PFF and assessed the internalization via live-cell imaging. There is endocytosis of the α-syn-pHrodo PFF in WT neurons, whereas the internalization of α-syn-pHrodo PFF was significantly reduced in $Aplp1^{-/-}$ neurons (Fig. 2a; Supplementary Fig. 1f). Overexpression of Aplp1 by lentivirus transduction significantly increased the endocytosis of α-syn-pHrodo PFF in WT neurons, and re-expression of Aplp1 in $Aplp1^{-/-}$ neurons restored the endocytosis of α-syn-pHrodo PFF in $Aplp1^{-/-}$ neurons (Fig. 2a; Supplementary Fig. 1f). Isolation of the endolysosomal fraction via differential centrifugation indicates that α-syn-biotin PFF treatment of cortical neurons led to an enrichment of α-syn-biotin PFF in the endosomal fraction in WT neurons, and a significant reduction in $Aplp1^{-/-}$ neurons (Fig. 2b, c). Lentiviral transduction of Aplp1 significantly enhanced the amount of α-syn-biotin PFF in the endolysosomal fraction of WT neurons and $Aplp1^{-/-}$ neurons (Fig. 2b, c). The purity of the endolysosome fraction has been validated with mitochondrial, nuclear, and cytosol markers (Supplementary Fig. 5d).

To further support that Aplp1 mediates the endocytosis of α-syn-biotin PFF into endosomes, the intensity of co-localized α-syn-biotin PFF with Rab7 (endosome marker) in WT and $Aplp1^{-/-}$ cortical neurons was measured. α-Syn-biotin PFF co-localized with Rab7 in WT neurons, whereas in $Aplp1^{-/-}$ neurons, there were less α-syn-biotin PFF co-localized with Rab7 (Fig. 2d; Supplementary Fig. 1g). Overexpression of Aplp1 in WT or $Aplp1^{-/-}$ neurons significantly increased the intensity of α-syn-biotin PFF co-localizing with Rab7 (Fig. 2d; Supplementary Fig. 1g).

While overexpression of Aplp1 increased the co-localization of α-syn-biotin PFF with Rab7, overexpression of App or Aplp2 did not increase the intensity of co-localization, compared to control neurons (Fig. 2e). Next, we studied the relationship between the deletion mutants of Aplp1 and the intensity of the co-localization of α-syn-biotin

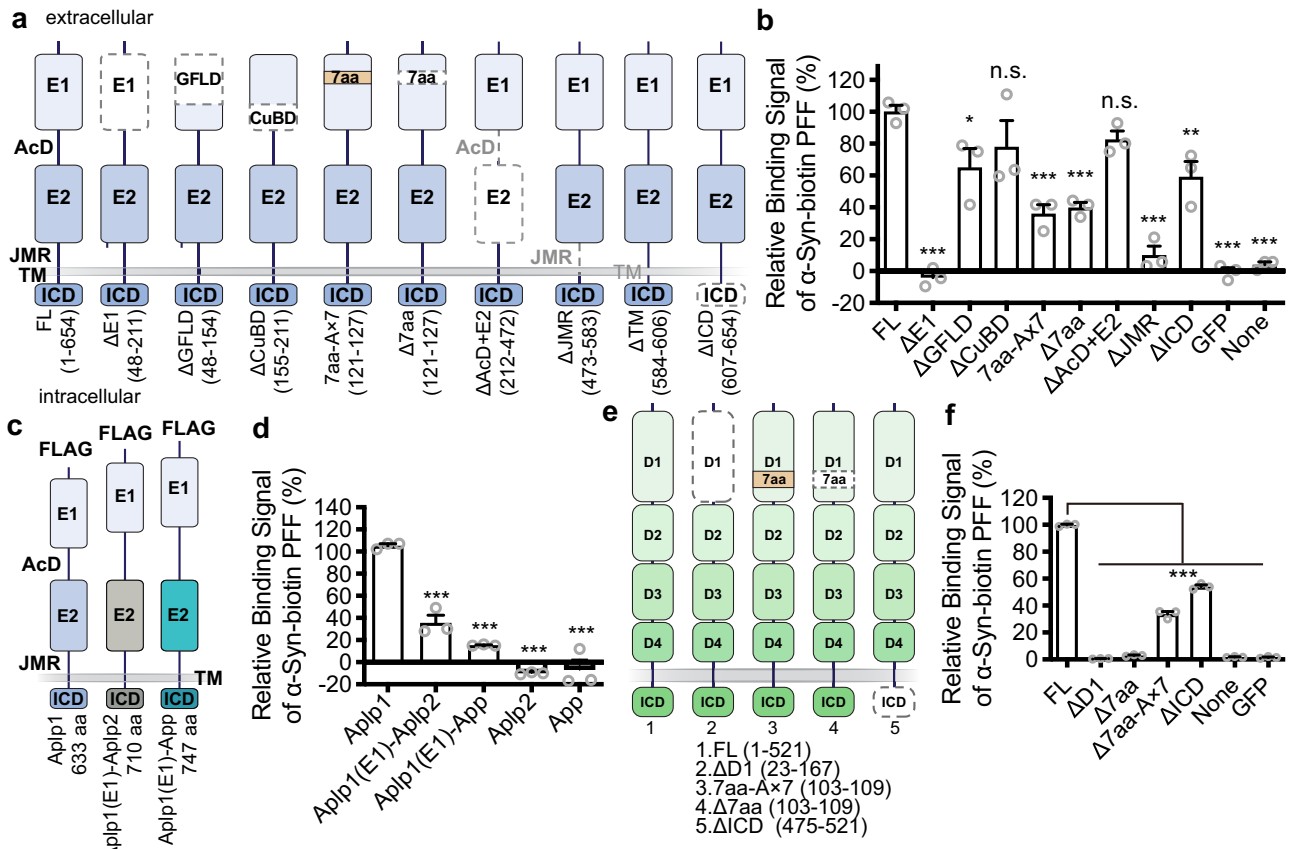

**Fig. 1 | α-Syn PFF binds to Aplp1. a** Schematic diagram of Aplp1 deletions mutants: the ΔE1, ΔAcD+E2, ΔJMR (juxtamembrane region), ΔTM (transmembrane region), ΔICD (intracellular domain), two subdomain deletion mutants of ectodomain E1 (ΔGFLD and ΔCuBD), and the deletion mutant (Δ7aa) and the substitutional mutant (7aa-A × 7) of a seven-amino acid (7aa) motif. **b** Quantification of binding signals of deletion mutants of Aplp1 with α-syn-biotin PFF by normalization with the expression of indicated deletion mutants of Aplp1. (*p*-values: FL vs. ΔE1 < 0.0001, FL vs. ΔGFLD 0.0267, FL vs. ΔCuBD 0.2771, FL vs. 7aa-A × 7 < 0.0001, FL vs. Δ7aa 0.0001, FL vs. ΔAcD + E2 0.5265, FL vs. ΔJMR < 0.0001, FL vs. ΔICD 0.0076, FL vs. GFP < 0.0001, FL vs. None <0.0001). **c, d** Schematic diagram of Aplp1 chimeras. Quantification of binding signals of Aplp1(E1)-Aplp2 and Aplp1(E1)-App chimeras with α-syn-biotin PFF by normalization with the expression of indicated Aplp1 chimeras. (*p*-values: Aplp1 vs. Aplp1(E1)-Aplp2 < 0.0001, Aplp1 vs. Aplp1(E1)-App <0.0001, Aplp1 vs. Aplp2 < 0.0001, Aplp1 vs. App <0.0001). **e** Schematic diagram of Lag3 and deletions mutants. **f** Quantification of binding signals of deletion mutants of Lag3 with α-syn-biotin PFF by normalization with the expression of indicated deletion mutants of Lag3. (*p*-values: FL vs. ΔD1 < 0.0001, FL vs. Δ7aa < 0.0001, FL vs. Δ7aa-A × 7 < 0.0001, FL vs. ΔICD < 0.0001, FL vs. None <0.0001, FL vs. GFP < 0.0001) **b, d, f** ****P* < 0.001, n.s. not significant. Data are the means ± SEM, from 3 individual experiments, one-way ANOVA followed by Dunnett's correction. Source data are provided as a Source Data file.

PFF with Rab7. The E1 domain deletion mutant significantly decreases the intensity of co-localization compared to full-length Aplp1 (Fig. 2e), the AcD + E2 domain and ICD deletion mutants do not significantly reduce the intensity of co-localization, compared to full-length Aplp1 (Fig. 2e). These data show that the E1 domain of Aplp1 is involved in the endocytosis of α-syn-biotin PFF, but not AcD + E2 or ICD. Furthermore, we investigated the endocytosis of α-syn-biotin PFF by transfecting the Aplp1 chimeras into neurons and found that both Aplp1(E1)-Aplp2 and Aplp1(E1)-App chimeras fail to increase the intensity of the co-localization (Fig. 2f). These results show that Aplp1 is involved in the endocytosis of α-syn-biotin PFF into neurons and that the E1 domain of Aplp1 is essential for endocytosis.

## Aplp1 induces α-syn pathology
We then asked whether knocking out Aplp1 prevents the pathology induced by α-syn PFF. We administered α-syn PFF or PBS, to WT and *Aplp1*[−/−] cortical neurons at 7 days in vitro and assessed α-syn pathology via phosphorylated serine 129 (pS129) of α-syn after ten days as published[33,34]. In WT neurons treated with α-syn PFF, the level of pS129 was significantly increased compared to WT neurons treated with PBS. In *Aplp1*[−/−] neurons treated with α-syn PFF, the level of pS129 was significantly reduced compared to WT neurons treated with α-syn PFF

(Fig. 2g, h). Overexpression of Aplp1 via lentiviral transduction significantly increased the level of pS129 in both WT neurons and *Aplp1*[−/−] neurons compared to lentiviral control virus transduction (Fig. 2g, h).

Two weeks after α-syn PFF was administered to WT and *Aplp1*[−/−] cortical neurons, we examined α-syn and pS129 levels from lysates sequentially extracted in 1% Triton X-100 (TX-soluble) and 2% SDS (TX-insoluble). In WT neuronal lysates, α-syn PFF led to substantial insoluble α-syn and pS129, whereas in *Aplp1*[−/−] neuronal lysates there was a significant reduction in insoluble α-syn and pS129 (Fig. 2i–k). Overexpression of Aplp1 via lentiviral transduction in WT and *Aplp1*[−/−] neurons increased the amount of insoluble α-syn and pS129 levels compared to lentiviral control virus transduction (Fig. 2i–k).

## Aplp1 and Lag3 bind to each other
Since α-syn PFF binds to both Aplp1 and Lag3[20], and both Aplp1 and Lag3 participate in the internalization of α-syn PFF, we hypothesized that Aplp1 and Lag3 bind to each other and work together to facilitate the uptake of α-syn PFF and the subsequent transmission of pathologic α-syn. To test this hypothesis, we investigated the interaction between Aplp1 and Lag3.

Co-Immunoprecipitation (co-IP) studies showed that Lag3 pulls down Aplp1 by anti-Lag3 410C9 immunoprecipitation in WT mouse

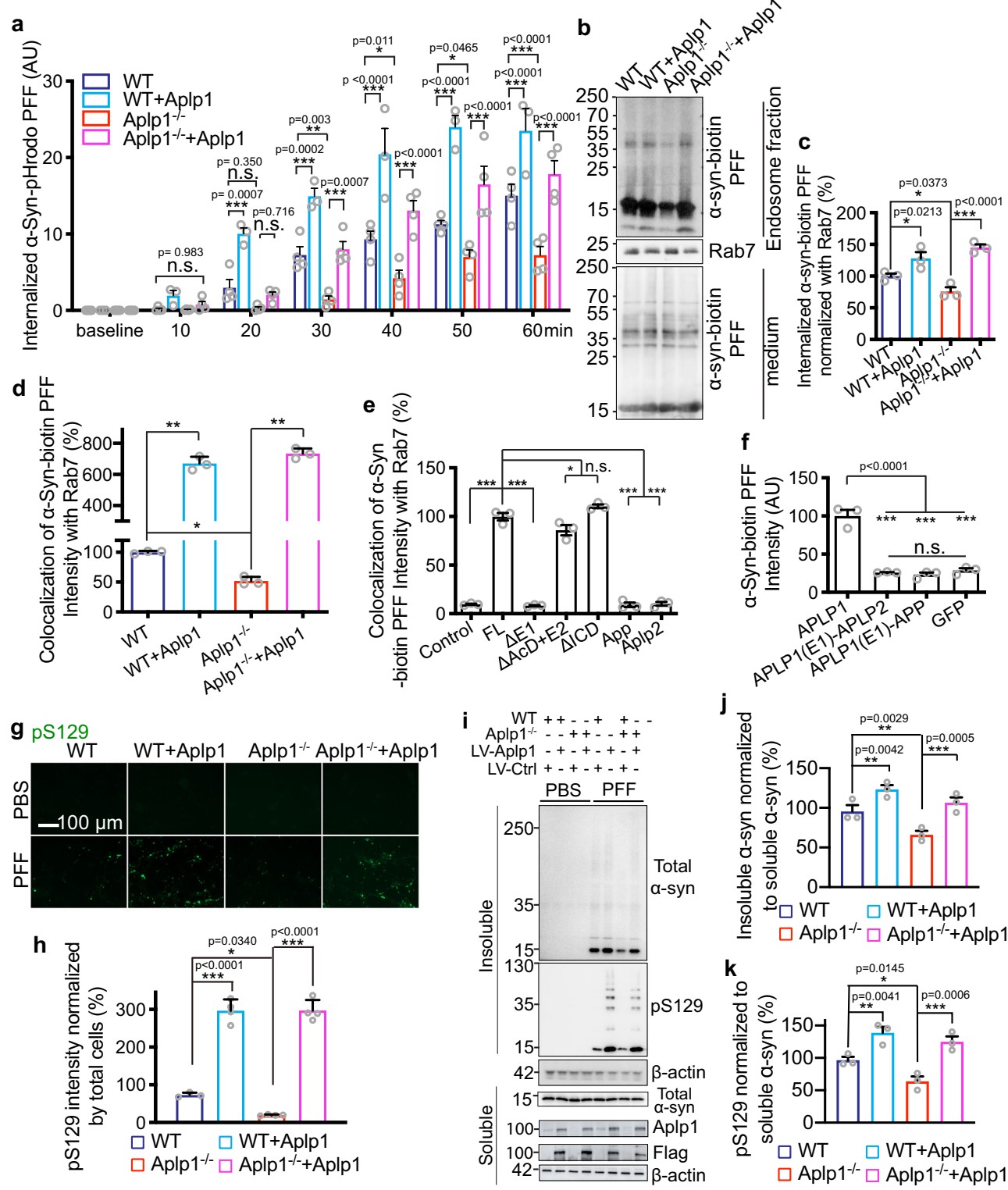

brain lysates, but not in *Lag3⁻/⁻* lysates (Fig. 3a). Conversely, Aplp1 pulls down Lag3 by anti-Aplp1 CT11 immunoprecipitation in WT mouse brain lysates, but not in *Aplp1⁻/⁻* lysates (Fig. 3b). To support that Lag3 is expressed in neurons, we determined the co-localization of anti-Lag3 and anti-NeuN in four brain regions of WT mice including the cortex (Supplementary Fig. 3c), the midbrain (Supplementary Fig. 3d), the thalamus (Supplementary Fig. 3e), and the hippocampus (Supplementary Fig. 3f). Immunostaining was not observed in *Lag3⁻/⁻* mice (Supplementary Fig. 3c–f). The quantifications showed that

approximately 50% of neurons are Lag3 positive. To substantiate the existence of Lag3 mRNA within neurons, we performed RNAScope in situ hybridization on the VMB and cortex areas in both WT and *Lag3⁻/⁻* mice. Our findings align with the Allen Brain Atlas and single-cell EMBL-EBI Expression Atlas In Mus Musculus, confirming that Lag3 expression is not only in WT microglia but also in VMB neurons (Supplementary Fig. 4a–c) and cortical neurons (Supplementary Fig. 4d–f). Notably, brain sections from *Lag3⁻/⁻* mice exhibited no discernible Lag3 signal.

**Fig. 2 | Aplp1 mediates the endocytosis of α-syn PFF and subsequent pathology.**
**a** Quantification of α-syn-pHrodo PFF endocytosis in WT and *Aplp1*−/− neurons control lentivirus (WT: 38 cells, Aplp1−/−: 40 cells) and Aplp1-lentivirus transduction (WT +Aplp1: 29 cells, *Aplp1*−/− + Aplp1: 35 cells). *n* = 4 (WT, *Aplp1*−/−, *Aplp1*−/− + Aplp1); *n* = 3 (WT + Aplp1) independent experiments. Two-way ANOVA with Tukey's correction. F (DFn, DFd): F (Time x Group) = F (18, 66) = 7.777; F (Time) = F (6, 66) = 143.2; F (Group) = F (3, 11) 46.15. **b**, **c** Immunoblot and quantification of α-syn-biotin PFF in the endolysosome fraction. *n* = 3 independent experiments. One-way ANOVA. **d** Quantification of the co-localization of α-syn-biotin PFF with Rab7 in WT and *Aplp1*−/− neurons. WT (38 cells), WT + Aplp1 (12 cells), *Aplp1*−/− (49 cells), *Aplp1*−/− + Aplp1 (13 cells) from 3 independent experiments. (*p*-values: WT vs. WT + Aplp1 0.0059, WT vs. *Aplp1*−/− 0.0230, *Aplp1*−/− vs. *Aplp1*−/− + Aplp1 0.0024). **e** Quantification of co-localization of α-syn-biotin PFF and Rab7 in WT neurons transiently expressing full-length Aplp1 (FL: 25 cells) and deletion mutants: ΔE1(10 cells), ΔAcD + E2 (12 cells), ΔICD (15 cells),

App (5 cells) and Aplp2 (5 cells), and control (16 cells) from 3 independent experiments. (*p*-values: FL vs. Control <0.0001, FL vs. ΔE1 < 0.0001, FL vs. ΔAcD + E2 0.0178, FL vs. ΔICD 0.0831, FL vs. App < 0.0001, FL vs. Aplp2 < 0.0001). **f** Quantification of co-localization signal of α-syn-biotin PFF in WT neurons transiently expressing Aplp1, chimeric Aplp1(E1)-Aplp2 and Aplp1(E1)-App. *n* = 3 independent experiments. **g** Immunostaining of anti-pS129 in WT and *Aplp1*−/− neurons expressing Aplp1 and control lentivirus, with α-syn PFF and PBS administration. Scale bar, 100 μm. **h** Quantification of (g). *n* = 3 (WT); *n* = 4 (WT + Aplp, *Aplp1*−/−, *Aplp1*−/− + Aplp1) independent experiments. **i–k**, Immunoblots for insoluble α-syn, pS129, soluble α-syn, and β-actin in WT and *Aplp1*−/− neuronal lysates post 15 days α-syn PFF treatment. *n* = 3 independent experiments. **d**, **e** One-way ANOVA with Dunnett's correction. **f**, **h**, **j**, **k** One-way ANOVA with Tukey's correction; **a**, **c**, **d**, **e**, **f**, **h**, **j**, **k** *P < 0.05, **P < 0.01, ***P < 0.001, n.s. not significant, Data are as means ± SEM. Source data are provided as a Source Data file.

We further assessed the cellular localization of Lag3 in a Lag3 Loxp reporter mouse line, with a YFP (yellow fluorescent protein) signal knocked into the Lag3 locus (*Lag3*L/L−YFP)[35]. In *Lag3*L/L−YFP mice, YFP marks all cells that express Lag3. Due to the nature of gene targeting, YFP is expressed in the cytosol of cells that normally express Lag3[35]. The immunoreactivity of anti-GFP, which recognizes YFP co-localized with the immunostaining of NeuN (Supplementary Fig. 5a) and MAP2 (Supplementary Fig. 5b) in the cortex. The lack of immunoreactivity of anti-GFP in WT mice indicates that the anti-GFP immunoreactivity is specific in *Lag3*L/L−YFP mice (Supplementary Fig. 5a, b).

Moreover, we cultured primary cortical neurons from WT and *Lag3*−/− mice and observed Lag3 expression in WT, but not in *Lag3*−/− neurons (Supplementary Fig. 5e). Aplp1 expression was observed in WT, but not in *Aplp1*−/− neurons (Supplementary Fig. 5f) confirming the specificity of the Lag3 and Aplp1 antibodies. The purity of primary cortical neurons was validated with neuronal, microglia, astrocyte, and oligodendrocyte markers (Supplementary Fig. 5e, f), with WT brain lysate as a positive control. Non-permeabilize neuron cultures revealed that Aplp1 and Lag3 co-localize on the surface of neurons as determined by wheat germ agglutinin (WGA, a plasma membrane marker) in WT primary cortical neurons[36–38], while there is no signal for Aplp1 and Lag3 in the double knockout of Aplp1 and Lag3 (*Aplp1*−/−/*Lag3*−/−) neurons supporting the specificity of the co-localization to the surface of neurons (Supplementary Fig. 5i). Taken together, the results of Aplp1 and Lag3 co-expression in neurons as well as published observations[20,31] indicate that both Aplp1 and Lag3 are expressed in neurons where they have the potential to interact.

To identify the Lag3-binding domain in Aplp1, we conducted co-immunoprecipitation experiments with the FLAG-Aplp1 deletion mutants described previously (Fig. 1a). The results showed that the E1 domain of FLAG-Aplp1 is involved in the interaction with Lag3-Myc (Fig. 3c). As Lag3-Myc binding to FLAG-Aplp1 is eliminated by deletion of the E1 domain of FLAG-Aplp1, we studied two additional deletion mutants of the E1 subdomain (ΔGFLD and ΔCuBD). The ΔGFLD mutant substantially reduced Lag3-Myc binding to FLAG-Aplp1, while the ΔCuBD mutant reduced binding to a lesser extent (Fig. 3d), suggesting that the GFLD subdomain in the E1 domain of Aplp1 (residues 48-154) is the major subdomain responsible for the Lag3 interaction. Similarly, the Aplp1-binding domain in Lag3 was determined via co-IP experiments. The D2 and D3 domains of Lag3-Myc are essential for the FLAG-Aplp1 interaction (Fig. 3e). Confirmation that the D2 and D3 deletion constructs of Lag3-Myc reach the cell membrane is provided in Fig. S1.

We next sought to investigate the structural basis for the interaction between the extracellular domains of APLP1 and Lag3, by using nuclear magnetic resonance (NMR) spectroscopy. Since our cellular data suggests that the E1 domain of Aplp1 binds to Lag3 via its D2 and D3 domains (Fig. 3c–e), we purified the recombinant E1 domain of APLP1 (A1E1), and D2 and D3 domains of Lag3 (L3D2 and L3D3) and performed NMR titration experiments to study their interaction. The

titration of L3D2 to A1E1 results in significant, but relatively small chemical shift deviations (CSDs, >0.01 ppm) of 37 residues of A1E1 (Fig. 3f, g), implying direct and weak binding of the two components. When the 37 residues were mapped to the structural model of A1E1 (Fig. 3h), we found that they are mainly located across the central β-sheet of A1E1. These results demonstrate that L3D2 directly interacts with A1E1 via binding to its β-sheet region. Of note, five residues (R90, V91, Y94, Q97, M98) located in the α-helix region of A1E1 (residue 90−98) also showed CSDs > 0.01 ppm. However, these residues face the inner side of the structure, implying that their CSDs may result from the conformational changes of its neighboring β-sheet, rather than the direct binding of L3D2 to the α-helix region of A1E1.

We further examined the interaction between L3D3 and A1E1 by NMR titration. In comparison to L3D2, titration of L3D3 to A1E1 resulted in CSDs (>0.01 ppm) of 5 residues of A1E1 (C84, L85, T111, I114, V135), suggesting that L3D3 is capable of binding to A1E1, but the binding affinity is lower than that of L3D2 to A1E1 (Supplementary Fig. 5c). Based on the NMR results, we substituted nine residues of APLP1 with alanine to generate FLAG-APLP1(mut9) and performed the co-IP experiment to assess the APLP1-Lag3 interaction (Fig. 3i). Co-IP results showed that the substitution significantly decreases the interaction between APLP1 and Lag3 (Fig. 3i). Taken together, these results demonstrate that A1E1 directly interacts with both D2 and D3 of Lag3, to form the Aplp1-Lag3 complex (Fig. 3j).

**Aplp1 and Lag3 dual role in binding and endocytosis of α-syn PFF**
Double knockout of Aplp1 and Lag3 (*Aplp1*−/−/*Lag3*−/−) mice were generated to determine the potential consequences of the interaction between Aplp1 and Lag3 on α-syn PFF-induced neurodegeneration (Supplementary Fig. 3a, b). α-Syn-biotin PFF-binding was reduced in both *Aplp1*−/− and *Lag3*−/− mice cortical neuron cultures (Supplementary Fig. 6a). α-Syn-biotin PFF-binding was reduced even further in *Aplp1*−/−/*Lag3*−/− cortical cultures (Supplementary Fig. 6a). Together Aplp1 and Lag3 account for greater than 40% of the binding of α-syn-biotin PFF to cortical neurons (Fig. 4a). *Aplp1*−/−/*Lag3*−/− cortical neurons were transduced with Aplp1, Lag3, or Aplp1 + Lag3 by lentivirus and α-syn-biotin PFF binding was assessed. Individual overexpression of Aplp1 or Lag3 leads to increased binding over baseline of α-syn-biotin PFF to *Aplp1*−/−/*Lag3*−/− cortical neurons (Fig. 4b; Supplementary Fig. 6b). Re-expression of Aplp1 and Lag3 together in *Aplp1*−/−/*Lag3*−/− cortical neurons led to a significant enhancement of binding that is greater than the sum of α-syn-biotin PFF binding to Aplp1 and Lag3 when individually expressed in *Aplp1*−/−/*Lag3*−/− cortical neurons (Fig. 4b; Supplementary Fig. 6b). Wild-type (WT) primary cortical neurons were transduced with Aplp1, Lag3, or Aplp1 + Lag3 by lentivirus for cell surface-binding assay. Overexpression of Aplp1 or Lag3 led to increased binding over the baseline of α-syn-biotin PFF to WT cortical neurons (Supplementary Fig. 7a, b). Overexpression of Aplp1 and Lag3 together led to a significant increase of binding compared to

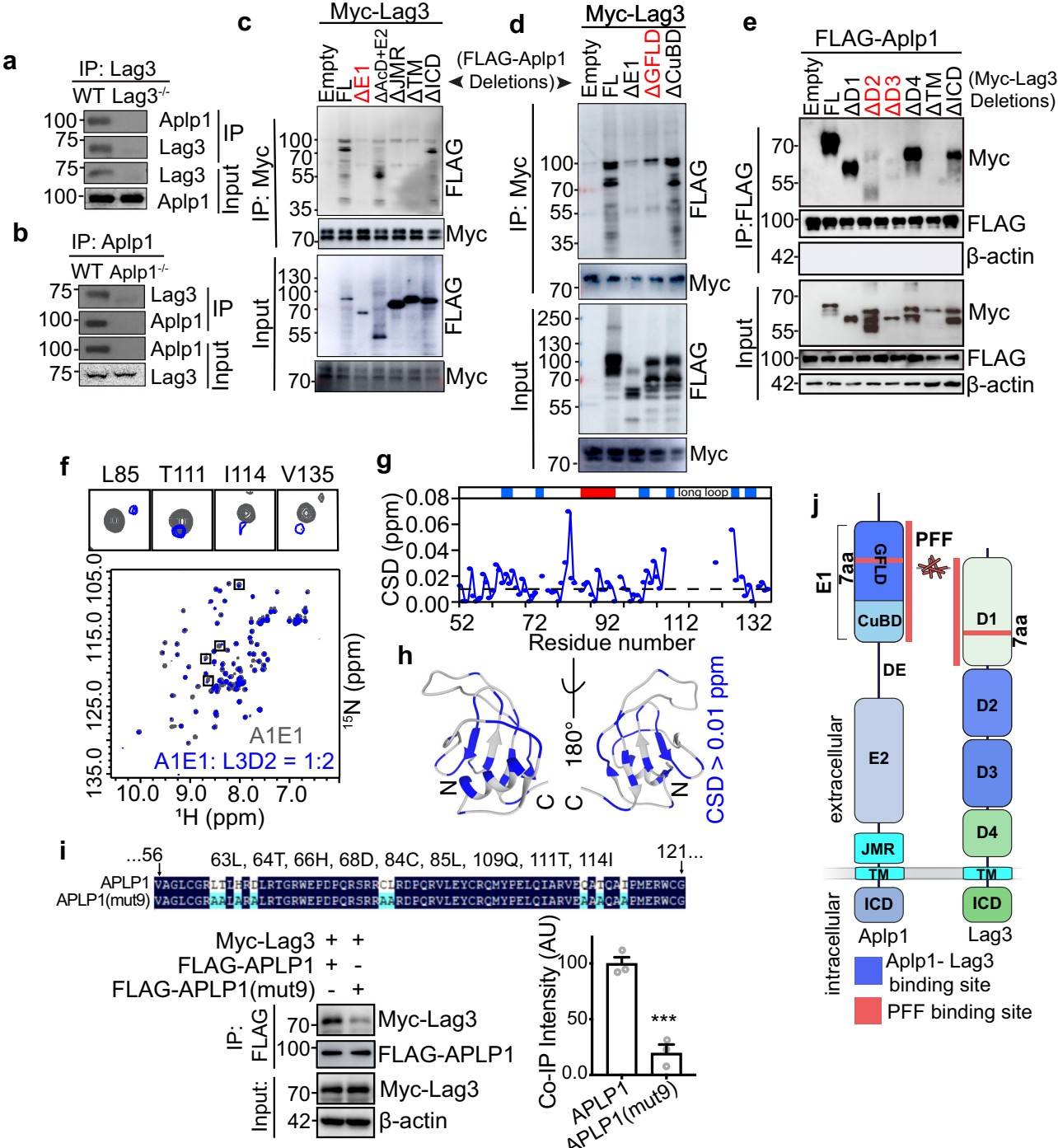

**Fig. 3 | Aplp1 and Lag3 bind to each other. a** Lag3 pulls down Aplp1 by anti-Lag3 410C9 immunoprecipitation in WT mouse brain lysates, but not in *Lag3*−/− lysates. **b** Aplp1 pulls down Lag3 by anti-Aplp1 CT11 immunoprecipitation in WT mouse brain lysates, but not in *Aplp1*−/− lysates. **c, d** Mapping of the Lag3-binding domains in Aplp1. HEK293FT cells were transfected with full-length (FL) or deletion mutants of FLAG-Aplp1, and Myc-Lag3 for co-IP experiments. The GFLD subdomain in the E1 domain of Aplp1 is the major subdomain responsible for the Lag3 interaction. **e** Mapping of the Aplp1-binding domains in Lag3. HEK293FT cells were transfected with FL, deletion mutants of Myc-Lag3, and FLAG-Aplp1 for co-IP experiments. Experiments in (**a**–**e**) were repeated three times independently with similar results. **f**–**h** Identification of the interface of A1E1 (E1 domain of APLP1) binding to L3D2 (D2 domain of LAG3). **f** Overlay of the 2D $^1$H-$^{15}$N HSQC spectra of A1E1 alone (gray) and in the presence of 2 molar folds of L3D2 (blue). Four residues with significant

calculated chemical shift deviations (CSDs) (> 0.03 ppm) are highlighted and enlarged in the black boxes. **g** Histogram of the CSDs of A1E1 in the presence of L3D2 at a molar ratio of 1:2 (A1E1/L3D2). The domain organization of A1E1 is indicated on the top, with blue boxes indicating the β-strands and the red box indicating the α-helix. A dashed line was drawn to highlight the residues with CSDs >0.01 ppm. **h** The 37 residues with large CSDs (>0.01 ppm) upon L3D2 titration are highlighted in blue on the ribbon diagram of the A1E1 modeled structure. **i** Validation of the NMR results. We substituted nine residues of APLP1 with alanine to generate FLAG-APLP1(mut9) and performed the co-IP experiment to assess the APLP1-Lag3 interaction. *n* = 3 independent experiments. Data are the means ± SEM, Two-tailed Student's *t*-test; *p*-value = 0.0009. **j** The scheme for the interaction among Aplp1, Lag3, and α-syn PFF. Source data are provided as a Source Data file.

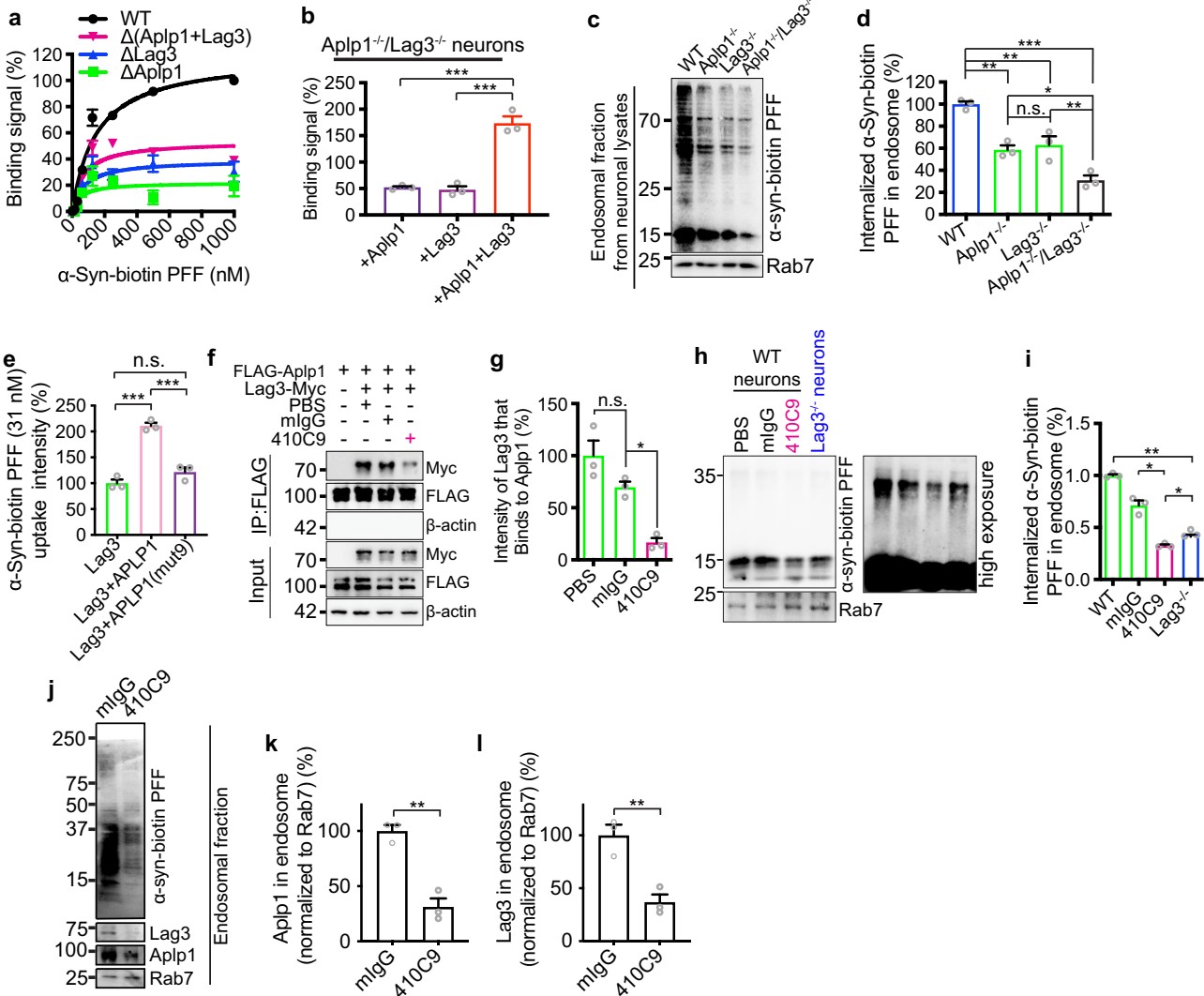

**Fig. 4 | The role of Aplp1-Lag3 interaction in mediating the binding, endocytosis of α-syn PFF, and subsequent pathology. a** Aplp1 and Lag3 account for more than 40% α-syn-biotin PFF binding to cortical neurons analyzed in supplementary Fig. 6a. Data are the means ± SD. **b** Co-expressing Aplp1 and Lag3 in *Aplp1⁻/⁻/Lag3⁻/⁻* cortical neurons substantially increases binding that is greater than the sum of α-syn-biotin PFF binding to Aplp1 and Lag3. Refer to supplementary Fig. 6b. (*p*-values: +Aplp1 vs. +Aplp1 + Lag3 0.0001, +Lag3 vs. +Aplp1 + Lag3 < 0.0001). **c, d** Immunoblots of endosomal fractions from WT, *Aplp1⁻/⁻*, *Lag3⁻/⁻* and *Aplp1⁻/⁻/Lag3⁻/⁻* cortical neurons. *Aplp1⁻/⁻/Lag3⁻/⁻* significantly decreases (-70%) internalized α-syn-biotin PFF compared to WT, *Aplp1⁻/⁻*, *Lag3⁻/⁻* neurons. (*p*-values: WT vs. *Aplp1⁻/⁻* 0.0019, WT vs. *Lag3⁻/⁻* 0.0039, WT vs. *Aplp1⁻/⁻/Lag3⁻/⁻* <0.0001, *Aplp1⁻/⁻* vs. *Aplp1⁻/⁻/Lag3⁻/⁻* 0.0208, *Lag3⁻/⁻* vs. *Aplp1⁻/⁻/Lag3⁻/⁻* 0.0091) **e**, APLP1 and Lag3 co-expression by transfection increases α-syn-biotin uptake in SH-SY5Y cells compared to APLP1(mut9) expression. The biotin signal was normalized using the intensity of Lag3. Lag3 (15 cells), Lag3 + Aplp1 (14 cells),

Lag3 + Aplp1(mut9) (11 cells). (*p*-values: Lag3 vs. Lag3 + Aplp1 < 0.0001, Lag3 vs. Lag3 + Aplp1(mut9) 0.1720, Lag3 vs. Lag3+Aplp1(mut9) 0.0003). **f, g** Anti-Lag3 410C9 significantly disrupts the co-IP of FLAG-Aplp1 and Myc-Lag3 in HEK293FT cells. (*p*-values: PBS vs. mIgG 0.1304, mIgG vs. 410C9 0.0159). **h, i** Anti-Lag3 410C9 (330 nM) reduced the internalization of α-syn-biotin PFF (1 μM) in WT neurons compared to *Lag3⁻/⁻* neurons. Quantification of the intensity of internalized α-syn-biotin PFF normalized by Rab7. (*p*-values: WT vs. *Lag3⁻/⁻* < 0.0057, mIgG vs. 410C9 < 0.0277, 410C9 vs. *Lag3⁻/⁻* 0.0152). **j–l** Both the levels of the endosomal Aplp1 and Lag3 were significantly decreased by 410C9 in WT cortical neuron cultures. *n* = 3 independent experiments. Two-tailed Student's *t*-test; *p*-value = 0.0019 (**k**), 0.0019 (**l**). For all experiments, *n* = 3 independent experiments. For graphs, **b, d, e, g, i, k, l** Data are the means ± SEM; **b, d, e, g, i** One-way ANOVA with Tukey's correction; *\**P* < 0.05, *\*\**P* < 0.01, *\*\*\**P* < 0.001; n.s. not significant. Source data are provided as a Source Data file.

Aplp1 or Lag3 alone consistent with binding in *Aplp1⁻/⁻/Lag3⁻/⁻* cortical neurons (Supplementary Fig. 7a, b).

We examined the internalization of α-syn-biotin PFF in the endolysosome-enriched fractions isolated from WT, *Aplp1⁻/⁻*, *Lag3⁻/⁻*, and *Aplp1⁻/⁻/Lag3⁻/⁻* cortical neurons. There was significantly less internalized α-syn-biotin PFF in *Aplp1⁻/⁻* or *Lag3⁻/⁻* neurons than in WT neurons (Fig. 4c, d). Knocking out both Aplp1 and Lag3 induced a significant decrease (-70%) in the amount of internalized α-syn-biotin PFF compared to WT neurons, which was also significantly less than the depletion of Aplp1 or Lag3 alone (Fig. 4c, d). Furthermore, we

administered α-syn-biotin PFF into Lag3-transfected cell cultures and found that Aplp1-Lag3 co-expression increased the uptake signal of α-syn-biotin PFF (Fig. 4e). Aplp1(mut9)-Lag3 co-expression failed to increase the uptake of α-syn-biotin PFF (Fig. 4e). Of note there is no appreciable difference in the signal of α-syn-biotin PFF uptake in Aplp1 versus Aplp1(mut9) transfected cells (Supplementary Fig. 6c) suggesting the Aplp1(mut9) itself does not appreciably interfere with α-syn-biotin PFF uptake. Furthermore, expression of APLP1(mut9) enhanced the uptake of α-syn-biotin PFF in Aplp1⁻/⁻ neurons (Supplementary Fig. 6h). We have determined that APLP1(mut9) significantly

reduced the interaction of APLP1 and Lag3 (Fig. 3i). Because Lag3(ΔD1) cannot bind to α-syn-biotin PFF binding[20], we co-expressed Lag3(ΔD1) with APLP1 or APLP1(mut9) and assessed the binding and uptake of α-syn-biotin PFF. The results showed that the APLP1-Lag3(ΔD1) significantly increased the binding and uptake of α-syn-biotin PFF (Supplementary Fig. 6k, l). In contrast, the APLP1(mut9)-Lag3(ΔD1) interaction significantly reduced the binding and uptake of α-syn-biotin PFF (Supplementary Fig. 6k, l). Taken together these results indicate the depletion of both Aplp1 and Lag3 significantly reduces the binding and the internalization of α-syn-biotin PFF, and Aplp1 and Lag3 form a complex that act as accessory proteins that significantly enhance the binding and uptake of α-syn-biotin PFF.

### The role of the Aplp1-Lag3 interaction in α-syn pathology, transmission, and neurotoxicity

To assess the role of the Aplp1-Lag3 interaction in α-syn pathology following α-syn PFF administration, α-syn PFF were incubated for 12 days in WT, *Aplp1*−/−, *Lag3*−/−, and *Aplp1*−/−/*Lag3*−/− primary cortical neurons. Depletion of both Aplp1 and Lag3 significantly decreased pS129 immunoreactivity by 70% which was induced by α-syn PFF compared to WT neurons. This reduction is significantly less than individual *Aplp1*−/− or *Lag3*−/− neurons (Fig. 5a, b). Furthermore, the levels of pS129 were assessed in *Aplp1*−/−/*Lag3*−/− neurons transduced with Aplp1, Lag3, or Aplp1 + Lag3 via lentivirus following α-syn PFF treatment. Re-expression of Aplp1 and Lag3 together in *Aplp1*−/−/*Lag3*−/− cortical neurons led to a significant enhancement of pS129 intensity that is larger than the sum of the individual contribution of the increased pS129 intensity when Aplp1 and Lag3 re-expressed individually in *Aplp1*−/−/*Lag3*−/− neurons (Fig. 5c, d).

A microfluidic neuronal culture device with three chambers, which are connected in tandem by two series of microgrooves (TCND1000, Xona Microfluidics) was used to examine the transmission of pathologic α-syn in WT, *Aplp1*−/−, *Lag3*−/−, and *Aplp1*−/−/*Lag3*−/− cortical cultures (Fig. 5e). Transmission of pathologic α-syn was monitored by pS129 immunoreactivity as previously described[20,34] (Fig. 5f). Four groups of neuron combinations were set up in successive chambers (1)-(2)-(3): (WT)-(WT)-(WT), (WT)-(*Aplp1*−/−)-(WT), (WT)-(*Lag3*−/−)-(WT), and (WT)-(*Aplp1*−/−/*Lag3*−/−)-(WT). There was an equivalent pS129 level in chamber 1 (C1) in *Aplp1*−/−, *Lag3*−/−, and *Aplp1*−/−/*Lag3*−/− cultures, which indicates that the pathologic α-syn levels in the WT neurons are at the same level at the initiation of the experiment (Fig. 5g). *Aplp1*−/−, *Lag3*−/−, and *Aplp1*−/−/*Lag3*−/− significantly blocked the transmission of pathologic α-syn to chamber 2 (C2) and chamber 3 (C3) (Fig. 5h, i), suggesting that both Aplp1 and Lag3 are required for the transmission of pathologic α-syn.

PI/Hoechst staining was used to assess cell death in cortical neuron cultures of WT, *Aplp1*−/−, *Lag3*−/−, and *Aplp1*−/−/*Lag3*−/− following administration of α-syn PFF. *Aplp1*−/−/*Lag3*−/− cultures exhibited significantly less cell death than *Aplp1*−/−, *Lag3*−/− or WT cultures (Fig. 5j, k).

### Roles of Aplp1 and the Aplp1-Lag3 in mediating α-syn PFF-induced neurodegeneration in vivo

To determine the roles of Aplp1 and the Aplp1-Lag3 complex contributions to neurodegeneration induced by α-syn PFF in vivo, we stereotactically injected the same amount of α-syn PFF (5 μg) into the right side of the dorsal striatum of *Aplp1*−/− and *Aplp1*−/−/*Lag3*−/−. As previously described in WT mice at 180 days after injection[15,20], there was a significant loss of dopamine (DA) neurons as detected by stereological counting of tyrosine hydroxylase (TH)- and Nissl-positive neurons in the substantia nigra pars compacta (SNpc) (Fig. 6a, b). There was a moderate rescue of DA neurons in α-syn PFF-injected *Aplp1*−/− mice (Fig. 6a, b). Depletion of App (*App*−/−) had no effect on DA neuron loss (Supplementary Fig. 8a, b) providing specificity to the role of Aplp1 in contributing to pathologic α-syn-induced neurodegeneration. In α-syn PFF-injected *Aplp1*−/−/*Lag3*−/− mice, there was extensive

preservation of DA neurons (Fig. 6a, b). In addition, although intrastriatal injection of α-syn PFF induced a decrease in striatal TH-positive immunoreactivity in WT mice, in *Aplp1*−/− mice there was the preservation of the striatal TH-positive immunoreactivity and in the *Aplp1*−/−/*Lag3*−/− mice there was significantly greater preservation of TH-positive immunoreactivity (Supplementary Fig. 8c, d). High-performance liquid chromatography-electrochemical detection (HPLC-ECD) analysis demonstrated a significant reduction in the levels of DA and its metabolites homovanillic acid (HVA), 3,4-dihydroxyphenylacetic acid (DOPAC), and 3MT in WT mice by α-syn PFF (Fig. 6c; Supplementary Fig. 8e–g). Depletion of Aplp1 mildly alleviated the DA deficits but the results are not significantly different (Fig. 6c; Supplementary Fig. 8e–g). The DA deficits were eliminated in the *Aplp1*−/−/*Lag3*−/− mice (Fig. 6c; Supplementary Fig. 8e–g). Immunoreactivity for pS129 was monitored in the SNpc TH-positive neurons 180 days after α-syn PFF injection. As previously described[15,20], we observed substantial pS129 immunostaining in WT SNpc TH-positive neurons (Fig. 6d, e). In contrast, in *Aplp1*−/− SNpc TH-positive neurons, pS129 immunostaining was significantly reduced by approximately 60%, and depletion of both Aplp1 and Lag3 (*Aplp1*−/−/*Lag3*−/− mice) significantly reduced the pS129 immunostaining by approximately 90% (Fig. 6d, e).

α-Syn transmission in vivo was monitored by assessing the distribution of pS129 α-syn pathology in the coronal brain sections of WT, *Aplp1*−/− and *Aplp1*−/−/*Lag3*−/− mice (Fig. 6f–h). We observed substantial immunoreactivity of pS129 in coronal brain sections of WT mice at 180 days after injection (Fig. 6f–h). In contrast, in *Aplp1*−/− mice, the immunoreactivity of pS129 was significantly reduced by approximately 60% and co-depletion of the Aplp1-Lag3 significantly blocked the immunoreactivity of pS129 by greater than 90% (Fig. 6f–h).

WT mice injected with α-syn PFF were significantly impaired in their performance on the pole test and the cylinder test (Fig. 6i, j). These behavioral deficits were significantly reduced in the *Aplp1*−/− mice and were eliminated in the *Aplp1*−/−/*Lag3*−/− mice (Fig. 6i, j). We do not observe obvious common features between *Lag3*−/− and *Aplp1*−/− mice nor are there any reports in the literature of common features[20,39,40].

### Role of Lag3 antibody in blocking α-syn PFF-induced neurodegeneration in vivo

Previously we showed that the Lag3 antibody, 410C9, significantly decreased the internalization of α-syn-biotin PFF in WT cortical neurons[20]. Accordingly, the domain of Lag3 that 410C9 recognizes was assessed to determine whether it could serve as a reagent to examine the role of the interaction of Lag3 with Aplp1 in the internalization of pathologic α-syn. We found that 410C9 recognizes the D3 domain of Lag3 (Supplementary Fig. 6d)[41]. Since Aplp1 interacts with the D3 domain of Lag3 (see Fig. 3e, l; Supplementary Fig. 5c), we assessed whether 410C9 could disrupt the Aplp1-Lag3 interaction. 410C9 significantly disrupts the co-IP of FLAG-Aplp1 and Lag3-Myc in HEK293FT cells (Fig. 4f, g). We have determined that 410C9 can inhibit the interaction between Lag3 and α-syn-biotin PFF[20] and has no effect on the interaction between Aplp1 and α-syn-biotin PFF (Supplementary Fig. 6g, i, j). Aplp1 can increase the binding and uptake of α-syn-biotin PFF in the presence of Lag3, and anti-Lag3 can significantly inhibit the binding and uptake of α-syn-biotin PFF in when Lag3 is expressed alone and when Lag3 is co-expressed with Aplp1 (Supplementary Fig. 6i, j).

α-Syn PFF treatment of the cellular extract of FLAG-Aplp1 and Lag3-Myc transfected HEK293FT cells significantly increased the co-IP of FLAG-Aplp1 and Lag3-Myc (Supplementary Fig. 6e, f). We compared the inhibitory effects of 410C9 on the internalization of α-syn-biotin PFF in WT versus *Lag3*−/− cortical neurons. In the endosomal fraction, 410C9 blocked the internalization of α-syn-biotin PFF significantly more in WT cultures than in *Lag3*−/− cultures (Fig. 4h, i). Both the levels

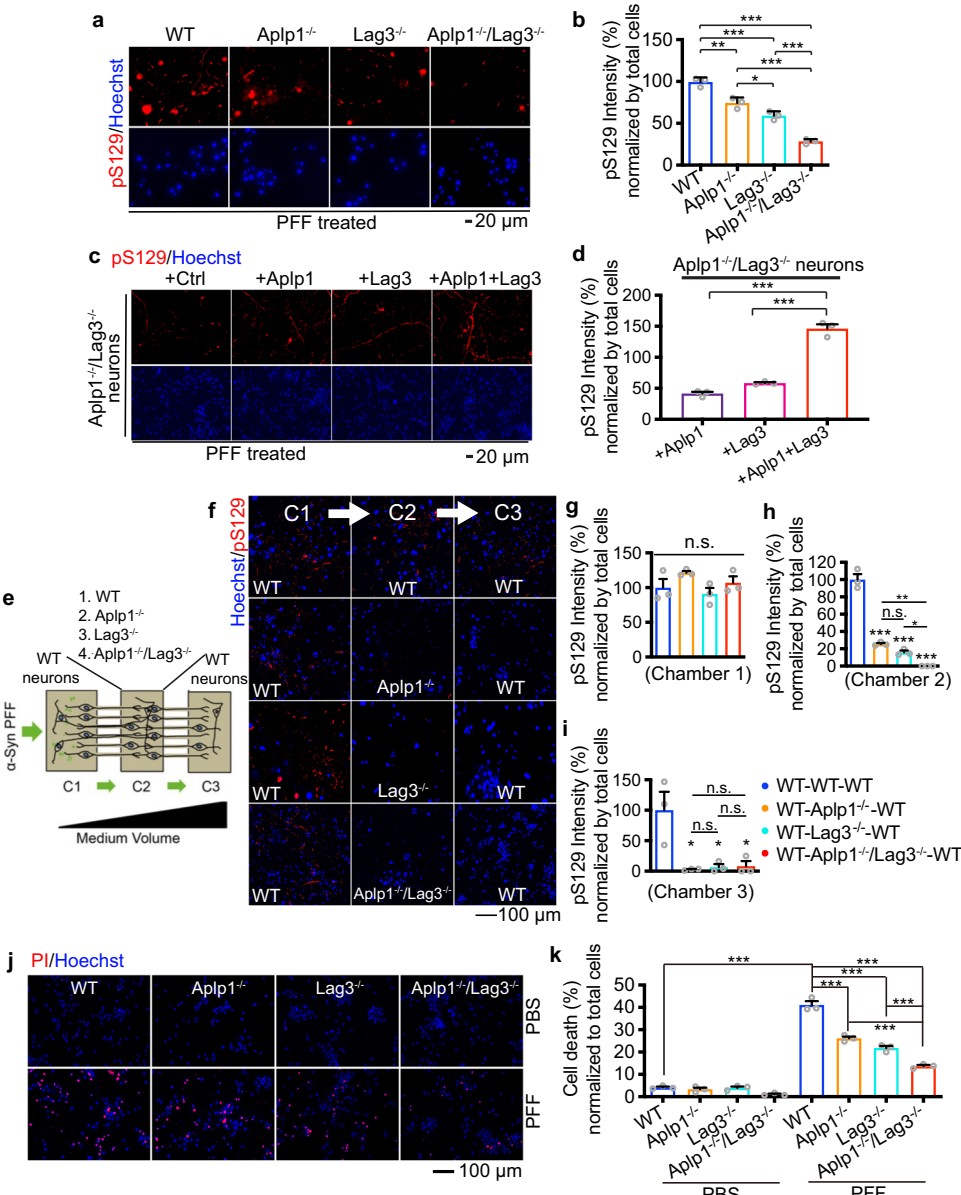

**Fig. 5 | The role of Aplp1-Lag3 interaction in α-syn pathology propagation, transmission, and neurotoxicity in vitro. a, b** Aplp1-Lag3 double deletion significantly decreased (70%) pS129 immunostaining induced by α-syn PFF, compared to *Aplp1⁻/⁻*, *Lag3⁻/⁻*, or WT neurons. Scale bar, 20 μm. (*p*-values: WT vs. *Aplp1⁻/⁻* 0.0016, WT vs. *Lag3⁻/⁻* < 0.0001, WT vs. *Aplp1⁻/⁻/Lag3⁻/⁻* <0.0001, *Aplp1⁻/⁻* vs. *Lag3⁻/⁻* 0.0296, *Aplp1⁻/⁻* vs. *Aplp1⁻/⁻/Lag3⁻/⁻* < 0.0001, *Lag3⁻/⁻* vs. *Aplp1⁻/⁻/Lag3⁻/⁻* 0.0004). **c, d** Immunostaining of anti-pS129 in *Aplp1⁻/⁻/Lag3⁻/⁻* neurons, treated with ºα-syn PFF, transduced with Aplp1, Lag3, or Aplp1 + Lag3. Scale bar, 20 μm. (*p*-values: +Aplp1 vs. +Aplp1 + Lag3 < 0.0001, +Lag3 vs. +Aplp1 + Lag3 < 0.0001). **e** Schematic of a microfluidic device with three chambers. α-Syn transmission from chamber 1 (C1) to chamber 2 (C2), to chamber 3 (C3) 14 days α-syn PFF treatment in C1. The different combinations of neurons tested in C2, listed as C1-(C2)-C3, are WT-(WT)-WT, WT-(*Aplp1⁻/⁻*)-WT, WT-(*Lag3⁻/⁻*)-WT, WT-(*Aplp1⁻/⁻/Lag3⁻/⁻*)-WT. **f–i** Immunostaining images and quantification of pS129 signals in the transmission. Scale bar, 100 μm. (*P*-values: Chamber 2(h), WT-WT-WT vs. WT-

*Aplp1⁻/⁻*-WT < 0.0001, WT-WT-WT vs. WT-*Lag3⁻/⁻*-WT < 0.0001, WT-WT-WT vs. WT-*Aplp1⁻/⁻/Lag3⁻/⁻*-WT < 0.0001, WT-*Aplp1⁻/⁻*-WT vs. WT-*Aplp1⁻/⁻/Lag3⁻/⁻*-WT 0.0032, WT-*Lag3⁻/⁻*-WT vs. WT-*Aplp1⁻/⁻/Lag3⁻/⁻*-WT 0.0427; Chamber 3(i) WT-WT-WT vs. WT-*Aplp1⁻/⁻*-WT 0.0111, WT-WT-WT vs. WT-*Lag3⁻/⁻*-WT 0.0141, WT-WT-WT vs. WT-*Aplp1⁻/⁻/Lag3⁻/⁻*-WT 0.0154). **j, k** PI/Hoechst staining for cell death in WT, *Aplp1⁻/⁻*, *Lag3⁻/⁻*, and *Aplp1⁻/⁻/Lag3⁻/⁻* cortical neuron cultures, treated with α-syn PFF. Scale bar, 100 μm. Aplp1-Lag3 deletion exhibited significantly less cell death than in *Aplp1⁻/⁻*, *Lag3⁻/⁻*, or WT cultures, treated with α-syn PFF. (*p*-values: PBS WT vs. PFF WT < 0.0001, PFF group: WT vs. *Aplp1⁻/⁻* < 0.0001, WT vs. *Lag3⁻/⁻* < 0.0001, WT vs. *Aplp1⁻/⁻/Lag3⁻/⁻* < 0.0001, *Aplp1⁻/⁻* vs. *Aplp1⁻/⁻/Lag3⁻/⁻* <0.0001, *Lag3⁻/⁻* vs. *Aplp1⁻/⁻/Lag3⁻/⁻* < 0.0001). For all experiments, *n* = 3 independent experiments. For graphs, Data are the means ± SEM. One-way ANOVA with Turkey's correction. **P* < 0.05, ***P* < 0.01, ****P* < 0.001, n.s. not significant. Source data are provided as a Source Data file.

of the endosomal Aplp1 and Lag3 were significantly decreased by 410C9 in WT cortical cultures (Fig. 4j–l). Taken together these results further support an Aplp1 interaction with Lag3 and suggest that 410C9 disrupts the Aplp1- Lag3 interaction leading to enhanced blockade of the internalization of α-syn PFF, which is greater than that observed in *Lag3⁻/⁻* cultures.

Next, we asked whether 410C9 could block the neurodegeneration induced by intrastriatal injection of α-syn PFF. WT mice were inoculated with α-syn PFF and then were treated with 410C9 or control mouse IgG (mIgG) (10 mg/kg, intraperitoneally [i.p.]), followed by body weight measurement and weekly treatment with 410C9 or mIgG for 180 days (a total of 26 treatments) (Supplementary Fig. 9a). To

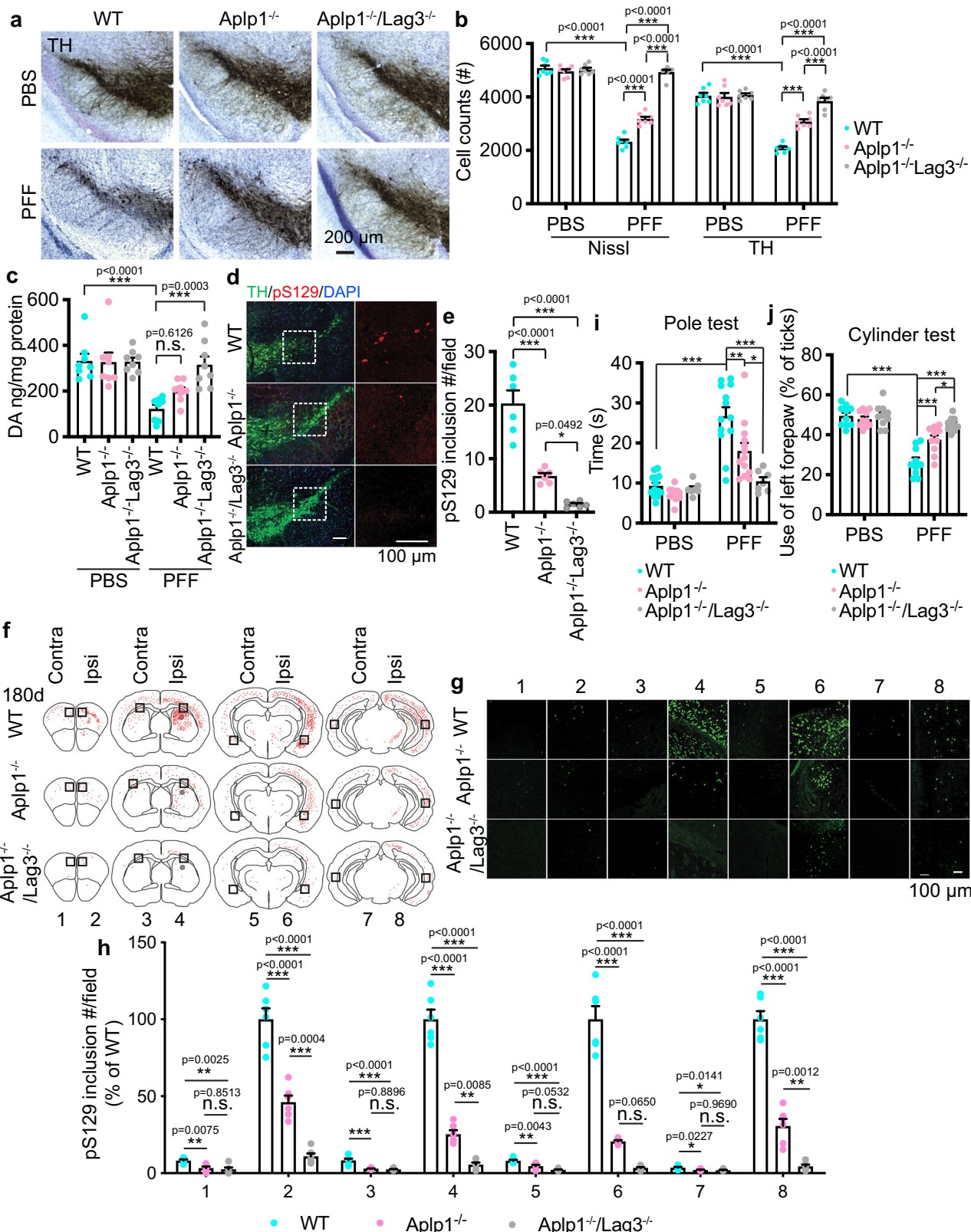

determine whether i.p. administration of 410C9 led to an appreciable antibody concentration within the brain, we measured 410C9 levels in the cerebral spinal fluid (CSF) at 3, 7, and 10 days following a single i.p. injection (10 mg/kg)[42]. Within 3 days following injection, the CSF levels of 410C9 reached 0.5% of the plasma levels (Supplementary Fig. 9b) similar to what has been reported with other antibodies that cross the blood-brain-barrier[43,44], and then persisted for up to 10 days

(Supplementary Fig. 9c). In contrast, the plasma levels of 410C9 significantly decreased at day 7, were not detectable at day 10 (Supplementary Fig. 9d).

To test the efficacy of 410C9 in inhibiting the internalization of α-syn PFF in neurons in vivo, WT mice were pre-treated with 410C9 or mIgG (10 mg/kg) for 3 days, followed by intrastriatal injection of α-syn-biotin PFF. We examined the co-localized intensity of α-syn-biotin PFF

**Fig. 6 | Roles of Aplp1 and the Aplp1-Lag3 interaction in mediating α-syn PFF-induced neurodegeneration in vivo. a** Representative TH (tyrosine hydroxylase) immunohistochemistry and Nissl staining images of dopamine (DA) neurons in the SNpc of α-syn PFF-injected hemisphere in the WT, *Aplp1⁻/⁻*, and *Aplp1⁻/⁻/Lag3⁻/⁻* mice. **b** Stereological counting of TH- and Nissl-positive neurons in the substantia nigra after 6 months of α-syn PFF injection (WT: $n = 7$; *Aplp1⁻/⁻*: $n = 7$; *Aplp1⁻/⁻/Lag3⁻/⁻*: $n = 7$). **c** DA concentrations in the striatum of PBS and α-syn PFF-injected mice measured at 180 days by HPLC. $n = 8$ mice per group, two-way ANOVA with Sidak's correction. **d, e** Representative images, and quantification of pS129 positive inclusions in the substantia nigra of WT, *Aplp1⁻/⁻*, and *Aplp1⁻/⁻/Lag3⁻/⁻* mice. ($n = 6$, each group). **f** Distribution of LB/LN-like pathology in the brain sections of α-syn PFF-injected WT, *Aplp1⁻/⁻*, and *Aplp1⁻/⁻/Lag3⁻/⁻* mice (pS129 positive neuron, red dots; pS129 positive neurites, red lines). **g, h** Representative images of LB/LN-like

pathology (the black box in **h**) and the quantification of pS129 intensity (green) from each coronal section (1–4) stained with pS129. ($n = 6$, each group). **i, j** Assessments of the behavioral deficits measured by the pole test (WT: $n = 13$; *Aplp1⁻/⁻*: $n = 13$; *Aplp1⁻/⁻/Lag3⁻/⁻*: $n = 7$) and cylinder test ($n = 10$, each group). ($p$-values: for pole test (**i**), PBS WT vs. PFF WT < 0.0001, PFF WT vs. PFF *Aplp1⁻/⁻* < 0.0010, PFF WT vs. PFF *Aplp1⁻/⁻Lag3⁻/⁻* < 0.0001, PFF *Aplp1⁻/⁻* vs. PFF *Aplp1⁻/⁻Lag3⁻/⁻* < 0.0312; For cylinder test (**j**): PBS WT vs. PFF WT < 0.0001, PFF WT vs. PFF *Aplp1⁻/⁻* 0.0004, PFF WT vs. PFF *Aplp1⁻/⁻Lag3⁻/⁻* < 0.0001, PFF *Aplp1⁻/⁻* vs. PFF *Aplp1⁻/⁻/Lag3⁻/⁻* 0.0304). For graphs, **e, h** One-way ANOVA with Tukey's correction; **b, j** Two-way ANOVA with Tukey's correction; Data are the means ± SEM, *$P < 0.05$. **$P < 0.01$. ***$P < 0.001$. n.s. not significant. Source data are provided as a Source Data file.

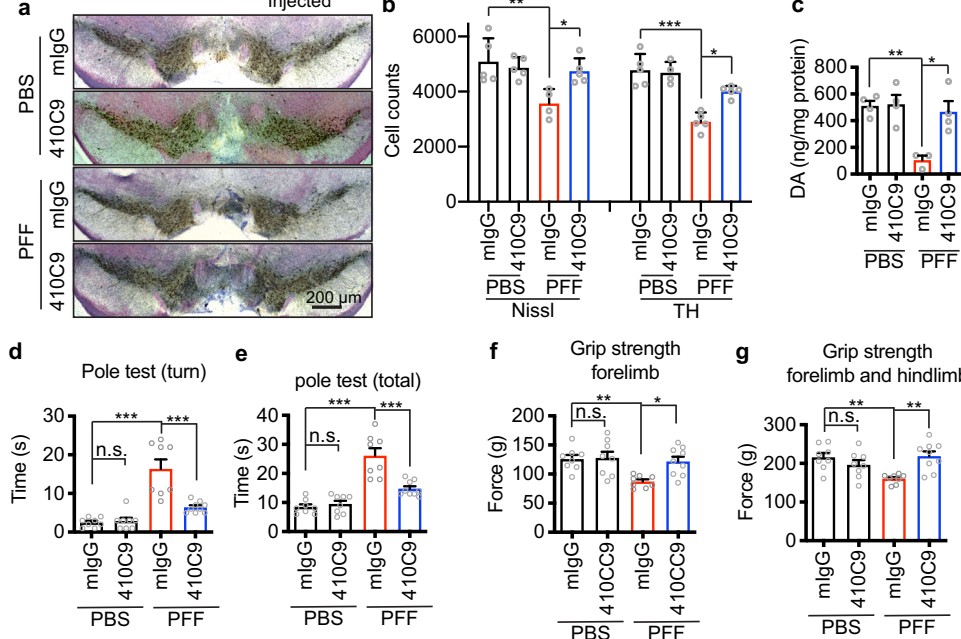

**Fig. 7 | The role of anti-Lag3 in blocking α-syn PFF-induced neurodegeneration in vivo. a** Representative images of TH and Nissl staining of SNpc DA neurons of WT mice treated with mIgG or 410C9 at 6 months after intrastriatal injection of α-syn PFF. **b** Stereology counts of data are the means ± SEM, (mIgG-PBS: $n = 5$; 410C9-PBS: $n = 5$; mIgG-PFF: $n = 4$; 410C9-PFF: $n = 5$). One-way ANOVA with Turkey's correction. ($p$-values: for Nissl: PBS mIgG vs. PFF mIgG 0.0022, PFF mIgG vs. PFF 410C9 0.0301, for TH: PBS mIgG vs. PFF mIgG <0.0001, PFF mIgG vs. PFF 410C9 0.0311) *$P < 0.05$, **$P < 0.01$, ***$P < 0.001$. **c** DA concentrations in the striatum of α-syn PFF or PBS-injected mice treated with 410C9 or mIgG measured at 6 months by HPLC-ECD (high-performance liquid chromatography-electrochemical detection). (mIgG-PBS: $n = 4$; 410C9-PBS: $n = 4$; mIgG-PFF: $n = 3$; 410C9-PFF: $n = 4$), Data are the means ± SEM, Two-way ANOVA with Tukey's correction; ($p$-values: PBS mIgG vs. PFF mIgG 0.0050, PFF mIgG vs. PFF 410C9 0.0111). **d–g** Behavioral deficits were

ameliorated by 410C9. Six months after α-syn PFF injection, the grip strength test and pole test were performed. Behavioral abnormalities in the grip strength test and pole test induced by α-syn PFF were ameliorated by 410C9 treatment (one day after stereotaxic injection). (mIgG-PBS: $n = 8$; 410C9-PBS: $n = 8$; mIgG-PFF: $n = 8$; 410C9-PFF: $n = 9$). Data are the means ± SEM, one-way ANOVA with Dunnett's correction; ($p$-values: **d** PBS mIgG vs. PBS 410C9 0.9970 PBS, mIgG vs. PFF mIgG 0.0001, PFF mIgG vs. PFF 410C9 0.0001; **e** PBS mIgG vs. PBS 410C9 0.9655 PBS, mIgG vs. PFF mIgG 0.0001, PFF mIgG vs. PFF 410C9 0.0001; **f** PBS mIgG vs. PBS 410C9 0.9986 PBS, mIgG vs. PFF mIgG 0.0089, PFF mIgG vs. PFF 410C9 0.0179; **g** PBS mIgG vs. PBS 410C9 0.5869 PBS, mIgG vs. PFF mIgG 0.0057, PFF mIgG vs. PFF 410C9 0.0026). *$P < 0.05$, **$P < 0.01$, ***$P < 0.001$. Source data are provided as a Source Data file.

with Rab7 in neurons with MAP2-positive staining and found that 410C9 significantly inhibited the co-localization of α-syn-biotin PFF with Rab7 in vivo (Supplementary Fig. 9e, f).

To test whether 410C9 retards disease progression (DA neuron loss, α-syn pathology, DA levels, and behavioral deficits), we initiated treatment with either 410C9 or mIgG in separate cohorts one-day post intrastriatal injection of α-syn PFF or PBS. At 180 days after α-syn PFF injection, stereological counting of TH- and Nissl-positive neurons in the SNpc revealed a significant loss of DA neurons in WT mice treated with mIgG (Fig. 7a, b). In contrast, 410C9 significantly reduced the loss of DA neurons induced by α-syn PFF (Fig. 7a, b).

In PFF-inoculated, mIgG-treated mice, pS129 α-syn was measured in three brain regions: the cortex, amygdala, and SNpc, primarily

ipsilateral to the injection site (Supplementary Fig. 9g–j). Importantly, 410C9 treatment significantly decreased the level of pS129 α-syn in these brain regions (Supplementary Fig. 9g–j). Thus, the simultaneous administration of 410C9 led to a reduction in α-syn PFF-induced pS129 α-syn pathology in vivo.

Immunoblot analysis demonstrated a significant decrease in TH and the DA transporter (DAT) in the PFF-injected WT mice treated with mIgG (Supplementary Fig. 9k–m), compared to the PBS-injected mice treated with mIgG. Importantly, 410C9 treatment significantly increased the expression of TH and DAT (Supplementary Fig. 9k–m).

The decrease of DA and its metabolites DOPAC, HVA, and 3MT in WT mice by α-syn PFF, can be prevented by 410C9, as detected by

HPLC-ECD (Fig. 7c; Supplementary Fig. 9n–p). Furthermore, the behavioral deficits by α-syn PFF in the pole test and grip strength test were alleviated by 410C9 (Fig. 7d–g).

Taken together these data indicate that Lag3 immunotherapy decreased the spread of pathologic α-syn in vivo, ameliorated motor dysfunction mediated by LB pathology, and rescued TH cell loss.

## Discussion

The major findings of the paper are that Aplp1 is a receptor for pathologic α-syn that facilitates pathologic α-syn transmission and that together with Lag3 it contributes to pathologic α-syn transmission and pathogenesis. Double knockout of Aplp1 and Lag3 and or an anti-Lag3 antibody that disrupts the Aplp1 and Lag3 interaction to almost completely block α-syn PFF-induced neurodegeneration and behavioral deficits. Aplp1 and Lag3 work as accessory proteins to mediate the uptake of pathologic α-syn whereas Aplp1 and Lag3 can mediate the uptake of pathologic α-syn independently, but together they enhance the uptake[45].

Emerging evidence suggests that misfolded α-syn binds to at least eight transmembrane proteins including heparan sulfate proteoglycans (HSPGs)[24,46], TLR2[23,25], neurexins[20,26,47], Na⁺/K⁺-ATPase subunit α3[26], Lag3[20], Aplp1[20], FcγRIIb[22], LRP1[48] and the cellular prion protein (PrPc)[27]. Of these α-syn binding proteins, HSPGs, Lag3, TLR2, FcγRIIb, and PrPc have been reported to be either directly or indirectly involved in the internalization of pathologic α-syn. LRP1 is reported to be involved in the uptake of both monomeric and oligomeric α-syn, but not fibrils[48] as reported here for Aplp1 and Lag3. Our findings indicate that Aplp1 plays a role in the internalization of pathologic α-syn in neurons and contributes to the spread and neurodegeneration induced by pathologic α-syn, which raised the question could Lag3 and Aplp1 interact. Aplp1 and Lag3 together account for greater than 40% of the neuronal binding and 70% of the neuronal uptake. In addition, the Aplp1-Lag3 accounts for the majority (>90%) of the pathologic spreading and toxicity induced by pathologic α-syn. Our findings that the Lag3 antibody, 410C9 disrupts the interaction of Aplp1 and Lag3 and protects against the degenerative process set in motion by pathologic α-syn that is equivalent to deleting both Aplp1 and Lag3 supports the notion that the Lag3-Aplp1 interaction is important in a degenerative process induced by cell-to-cell spread of pathologic α-syn. Thus, other α-syn binding proteins likely either play additional accessory roles in the endocytosis and toxicity of pathologic α-syn or are involved in other roles. For instance, pathologic α-syn binding to TLR2 is involved in the microglia response to pathologic α-syn[23,25].

Although Lag3 is an important protein in the immune system and its mRNA is enriched in the brain[49] and microglia[50–52], we confirmed that Lag3 is expressed in neurons through the comparison between WT and Lag3⁻/⁻ mice, and the use of Loxp reporter line with a YFP (yellow fluorescent protein) signal knocked into the Lag3 locus (Lag3^{L/L–YFP})[35]. Recently it has been suggested that Lag3 is not expressed in neurons and that it does not play a role in neurodegeneration induced by pathologic α-syn as the absence of Lag3 had no effect on neurodegeneration induced by overexpression of human A53T α-syn[53]. This contrasts with other reports that Lag3 is expressed in neurons[49] and that the absence of Lag3 significantly reduces the neurodegeneration induced by overexpression of human A53T α-syn[39]. Moreover, public databases indicate that mRNA for Lag3 is detectable in neurons in multiple species, albeit at low levels. The first is the Allen Brain databases for mouse brain, adult human brain, developing human brain, and non-primate brain[54], the GTEX portal[55,56], the BIOGPS portal for mouse, and human[57,58]. In addition, according to the EMBL-EBI single-cell expression atlas, in Mus Musculus, Lag3 mRNA is expressed at low to medium levels in oligodendrocyte precursor cells, newly formed oligodendrocytes, neurons, and microglia among other cell types[59,60]. These expression

data coupled with our immunohistochemistry, immunoblot analysis, RNAscope assay in WT mice compared to Lag3⁻/⁻ mice and the use of Loxp reporter line with a YFP (yellow fluorescent protein) signal knocked into the Lag3 locus (Lag3^{L/L–YFP}) provides us with a very high degree of confidence that Lag3 is expressed in neurons.

Aplp1 mRNA expression is enriched in oligodendrocytes[61], while Aplp1 protein has been reported to be restricted to the neuronal surface[62–65]. Aplp1's and Lag3's localization to neurons supports the notion that these two proteins cooperate in the internalization of pathologic α-syn in neurons. Since Lag3 message is enriched in microglia and Aplp1 mRNA is enriched in oligodendrocytes, it will be important to determine the role of Lag3 and Aplp1 in microglia and oligodendrocytes in pathologic α-syn toxicity.

Consistent with our co-immunoprecipitation experiments, NMR analysis confirmed that Aplp1 and Lag3 interact with each other where the E1 domain of Aplp1 binds to Lag3 via its D2 and D3 domains. Both Lag3 and Aplp1 exhibit a preference for binding α-syn in its amyloid state over its monomeric form, thereby facilitating cell-to-cell transmission[66]. Previous studies have highlighted that the Lag3 D1 and APLP1 E1 domains bind to α-syn through their alkaline surface. Furthermore, both proteins are localized on the cell surface, and recent findings suggest that APLP1 may interact with α-synuclein fibrils through electrostatic interactions akin to Lag3[66]. The significantly high chemical shift deviation of the Lag3 D2 domain and Aplp1 E1 domain indicates a direct interaction between Lag3 and Aplp1. We have determined that both Aplp1 and Lag3 utilize a positively charged surface to directly bind with the acidic C terminus of α-syn PFF[66]. α-Syn PFF binds to a common seven-amino acid stretch that is contained within the E1, GFLD subdomain of Aplp1 and the D1 domain of Lag3 raising the possibility that there might be a common structural motif that accounts for the binding of pathologic α-syn to Aplp1 and Lag3. Addressing these key questions may facilitate the optimization of Aplp1 and Lag3 targeted therapies aimed at pathologic α-syn cell-to-cell transmission.

### Limitations of the study

The lack of obvious common features between Lag3⁻/⁻ and Aplp1⁻/⁻ mice[20,39,40,67] and the observation that the absence of Lag3 does not significantly change the expression of Aplp1 and the absence of Aplp1 does not significantly change the expression of Lag3 could suggest the interaction reported here is non-physiologic or only pathologic. However, an interaction between two proteins does not necessarily impact their expression or localization levels. For instance, the interaction between G protein-coupled receptors (GPCRs) and β-arrestins, despite possessing distinct structural features, β-arrestins function as adapters for GPCR desensitization and internalization without directly affecting their localization or G protein signaling functions[68,69]. The physiological role of Aplp1 and Lag3 needs additional investigation. It is also possible that Aplp1 could promote Lag3 action through a different mechanism than a direct interaction, which requires further study. In addition, it will be intriguing to investigate if Aplp1 could also act as an accessory protein for other reported α-syn receptors. While our study mainly focused on the neuronal function of Lag3-Aplp1, it will be important to determine if these proteins also work closely in other cell types including in microglia, astrocytes, and oligodendrocytes.

## Methods

This research complies with all relevant ethical regulations. The animal studies were approved by the Johns Hopkins University Animal Care and Use Committee (ACUC). All animal studies were performed according to the NIH Guide for the Care and Use of Experimental Animals and the guidelines of the Institutional Animal Care Committee of Johns Hopkins University.

## Animals

C57BL/6 WT (Strain #:000664, *RRID:IMSR_JAX:000664*) were obtained from the Jackson Laboratories (Bar Harbor, ME). *Lag3*[-/-] mice[70] were obtained from Dr. Charles G. Drake when he was at Johns Hopkins University. *Aplp1*[-/-] mice[41] were obtained from Dr. Ulrike Muller at the University of Heidelberg. Both *Aplp1*[-/-] and *Lag3*[-/-] mice were kept in C57BL/6 background[31,40,67,70]. Double knockout of Aplp1 and Lag3 (*Aplp1*[-/-]/*Lag3*[-/-]) were generated by two consecutive crosses. *Aplp1*[-/-] and *Lag3*[-/-] were intercrossed to obtain the *Aplp1*[+/-]/*Lag3*[+/-] mice. These animals were further intercrossed (*Aplp1*[+/-]/*Lag3*[+/-] × *Aplp1*[+/-]/*Lag3*[+/-]) to obtain double knockout in the next generation (e.g., 6.25% *Aplp1*[+/+]/*Lag3*[+/+](WT) and 6.25% *Aplp1*[-/-]/*Lag3*[-/-]). All housing, breeding, and procedures were performed according to the guidelines of the NIH and Institutional Animal Care Committee of Johns Hopkins University for the Care and Use of Experimental Animals. All mice were housed under standard conditions of constant temperature of (22 ± 1 °C), relative humidity of 42%, and 12 h light cycle with food and water.

## Generation of α-syn monomer and PFF

Recombinant α-syn proteins were purified from ClearColi™ BL21 competent E. coli (Lucigen Corporation, USA, Cat# 60810-1) transformed with full-length α-syn in pRK172 vector per the following protocol dx.doi.org/10.17504/protocols.io.dm6gpbw28lzp/v1. The bacterial endotoxins were removed by Toxineraser endotoxin removal kit (GenScript Biotech Corp., USA) followed by the measurement of the level of endotoxin by ToxinSensor Chromogenic LAL Endotoxin Assay Kit (GenScript Biotech Corp., USA). The recombinant α-syn solution was aliquoted before fibrillization and stored at −80 °C until use. Before fibrillization, recombinant α-syn solution was centrifuged at 4 °C (12,000 × *g*, 15 min). Supernatant α-syn (5 mg/ml) was transferred into endotoxin-free Eppendorf tubes, and fibrils were generated by shaking for 5–7 days at 37 °C with 1000 RPM Eppendorf Thermomixer. α-Syn PFF were generated by sonication of fibrils with 20% amplitude for a total of 60 pulses (0.5 s on/off cycle). The fibrils and PFF were validated by Thioflavin T assay, transmission electron microscopy (TEM), immunoreactivity of anti-pS129 (1:1000 dilution, Abcam, Cat# ab51253, RRID:AB_869973) and neurotoxicity in primary neuronal culture.

## Generation of synthetic β-amyloid (1-42) fibrils and PFF

β-amyloid peptide was purchased from Anaspec (AS-23523-05). The β-amyloid peptide was dissolved clearly in Hexafluoro-2-propanol (HFIP), and kept in SpeedVac (Thermo Scientific, USA) for?? hours until HFIP completely dried down. The β-amyloid film was freshly suspended in DMSO at 2.2 mM and diluted in PBS to obtain a 250 µM stock solution. The β-amyloid fibrils were generated by incubation of the β-amyloid peptide solution at 37 °C for 24 h. The β-amyloid PFF were sonicated with 20% amplitude for a total of 60 pulses (0.5 s on/off cycle).

## Transmission electron microscopy (TEM) measurements

Protein samples (~100 ng/µL) were adsorbed to 400 mesh carbon-coated copper grids for 5 min, followed with three times rinse by Tris-HCl (50 mM, pH 7.4). The grids were then floated upon two consecutive drops of 2% uranyl formate and were dried with filter paper for imaging on a Phillips CM 120 TEM.

## Cell surface-binding assays and ImageJ analysis

SH-SY5Y cells (*RRID:CVCL_0019*) were transfected with related receptors (e.g., Aplp1, Lag3) or mutants. Two days after transfection, the cell cultures with the equivalent cell density were incubated with α-syn-biotin PFF for 1.5–2 h with different concentrations as indicated in DMEM media with 10% fetus bovine serum (FBS) at room temperature. After incubation, the cells were washed with DMEM media (3 times 5 min) and then fixed with 4% paraformaldehyde (PFA) in PBS for 15 min. After another 3 times wash with PBS, the cells were blocked for 30 min with 10% goat serum and 0.1 % Triton X-100 in PBS. Using alkaline-phosphatase-conjugated streptavidin (1:2000) in PBS supplemented with 5% goat serum and 0.05% Triton X-100, the cells were incubated overnight (~16 h). The bound streptavidin-alkaline phosphatase was visualized by 5-bromo-4-chloro-3-indolyl phosphatase/nitro blue tetrazolium reaction. Bound α-syn-biotin PFF to receptors-transfected SH-SY5Y cells was quantified with ImageJ (https://imagej.net/; *RRID:SCR_003070*) by normalizing with total cell numbers. Threshold was selected under Image/Adjust in order to achieve a desired range of intensity values for each experiment or image. All the images in each experiment were applied with the same threshold, to exclude the background and obtain the signal. After the exclusion of the background, the selected area in the signal intensity range of the threshold was measured using the measurement option under the Analyze/Measure menu. The area values with different α-syn-biotin (monomer or PFF) were input into the Prism program to obtain $K_d$ or $B_{max}$.

## Primary neuronal cultures, α-syn (-biotin) PFF and/or lentivirus transduction, and neuron binding assay

C57BL/6 mice were obtained from the Jackson Laboratories (Bar Harbor, ME). Primary cortical neurons were prepared from E15.5 mouse embryos and cultured in Neurobasal media supplemented with B-27, 0.5 mM L-glutamine, penicillin, and streptomycin (Invitrogen, USA) on tissue culture plates coated with poly-L-ornithine. The 3–4 days in vitro neuronal cultures were inhibited by F0503-5FU and were maintained by adding medium every 3 days. Lentiviral cFUGW vectors and particles were generated as previously described[20,71]. At 4 days in vitro, primary neurons were infected by lentivirus carrying related receptors, or empty vectors as a control [1 × 10⁹ transduction units (TU)/mL] for 72 h. α-Syn PFF (final concentration 5 µg/mL) were added at 7 days in vitro primary neurons and were incubated for 10–21 days followed by biochemical experiments or toxicity assays. Neurons were harvested for immunofluorescence staining or sequential extraction with Triton X-100 and SDS. Each experiment was performed in duplicate and repeated 3–6 times. To determine the bound signal, α-syn-biotin PFF with different concentrations was administered into primary neuron cultures with the equivalent cell density. Quantification of bound α-syn-biotin PFF to neurons was analyzed with ImageJ by normalizing with total cell numbers.

## Plasmids and deletion mutants

Lag3 cDNA clones were kindly obtained from Dr. Charles Drake at the Johns Hopkins University, School of Medicine. pcDNA3.1-Aplp1, App and Aplp2 cDNA clones were obtained from Dr. Yasushi Shimoda at Nagaoka University of Technology, and Dr. Gopal Thinakaran at the University of Chicago, and Dr. Ulrike Muller at the University of Heidelberg. Lag3 deletion mutants with a Myc tag, Aplp1 deletion mutants with a FLAG tag, 7aa substitution mutants of Lag3-Myc and Flag-Aplp1, and Aplp1 chimera mutants with a FLAG tag, were constructed by a ligation or In-Fusion method. Briefly, primers were designed to flank the sequences to be deleted or modified. DNA was PCR amplified with herculase polymerase (Agilent Technologies, USA) or CloneAmp HiFi PCR Premix (Clontech, USA). Amplicons were separated on a 1–2% agarose gel and appropriate bands were excised and isolated using a gel extraction kit (Qiagen, USA). These fragments were inserted into the pcDNA3.1-based plasmid or cFUGW lentivirus vector (Addgene cat #: 14883; RRID:Addgene_14883) using the T4 ligase (New England Biolabs, USA) or in-Fusion HD cloning kit (Clontech, USA). Vectors were transformed into Stellar competent cells (Clontech, USA, Cat# 636763) or chemically competent Oneshot™ Stbl3 E.Coli cells (Invitrogen, USA, Cat# C737303). The integrity of the constructs was

verified by sequencing. A list of oligonucleotides used is provided in Supplementary Table 1.

Genes encoding human A1E1 (residues 50–146 of *APLP1*), mouse L3D2 (residues 168–256), and L3D3 (residues 257–246) of *Lag3* were synthesized by Union-Biotech (Shanghai) Co, Ltd. Receptor genes were cloned into pET-28a (+), which contains an N-terminal his6-tag for purification.

Full-length human *APLP1* gene with a Flag tag in the N-terminal was cloned from pCAX *APLP1* (Addgene plasmid#30141) and the primers were shown in Key Resources Table (NheI-Flg-*hAPLP1*-F/XhoI-His-*hAPLP1*-R). Based on the NMR results, we substituted nine residues of *APLP1* with alanine: 63L, 64T, 66H, 68D, 84C, 85L, 109Q, 111T, and 114I, and obtained the human FLAG-*APLP1*(mut9) gene with a Flag tag in the N-terminal synthesized by GENEWIZ Bio. Inc. (Suzhou, China). Both pcDNA3.1-h*APLP1* and pcDNA3.1-h*APLP1*(mut9) genes were then ligated into the pcDNA3.1 backbone (Invitrogen, Cat# V79020) with *Nhe* I and *Xho* I sites to construct the plasmids.

Cell membrane localization of Aplp1 and Lag3 mutants: After PBS wash (three times within 5 min), SH-SY5Y cells were incubated for 10 min in 5 μg/mL Wheat Germ Agglutinin, Alexa Fluor™ 488 Conjugate (Invitrogen, W11261) mixed in HBSS. When labeling is complete, the cells were washed in HBSS. For cell membrane localization of Aplp1 and Lag3 mutants, the cells were permeabilized in 0.2% Triton X-100 for 10 min at room temperature then incubated with anti-FLAG 1:1000 (Cell Signaling Technology, Cat# 14793, RRID:AB_2572291) and anti-Myc 1:1000 (Cell Signaling Technology, Cat# 2276s, RRID:AB_331783) for 16 h at 4 °C, then washed and incubated with the secondary antibody.

## Protein expression, purification, and refolding

The three constructs encoding A1E1 (residues 50–146), L3D2 (residues 168–256), and L3D3 (residues 257–346) were transformed into *E. coli* BL21 (DE3) cells (Novagen, Cat# 69450), respectively. A similar protocol for gene expression, protein purification, and refolding was used for A1E1, L3D2, and L3D3. Cells were grown to $OD_{600} = 1.6$ and then induced by 1 mM IPTG. After shaking at 25 °C for 12 h, cells were harvested by centrifugation ($5053 \times g$, 16 min) and resuspended in a buffer containing 50 mM Tris, 500 mM NaCl at pH 8.0, then lysed by a high-pressure homogenizer. The lysate was centrifuged ($27, 216 \times g$, 30 min) at 4 °C and the pellet was washed in a buffer containing 50 mM Tris, 0.5% Triton X-100 at pH 8.0 and followed by another wash buffer of 50 mM Tris, 1 M NaCl. pH 8.0. Then the inclusion bodies were solubilized in 6 M guanidine hydrochloride containing 50 mM NaCl and 50 mM Tris, pH 8.0. The supernatant was loaded onto a 5-mL Ni-NTA column (GE Healthcare) and eluted with the buffer of 50 mM $Na_2HPO_4$ and 6 M guanidine hydrochloride, pH 4.0. The target protein was further purified by a 300SB-C3 RP-HPLC column (Agilent) with a linear gradient of acetonitrile (0–100%) and then lyophilized.

For the $^{15}$N-labeled A1E1 protein, the sample preparation protocol was the same as that for unlabeled A1E1 except that the cells over-expressing A1E1 proteins were grown in M9 minimal medium with $^{15}NH_4Cl$ (1 g/L).

Dry protein powder was solved in a buffer of 50 mM $Na_2HPO_4$, 50 mM NaCl, and 6 M Guanidine hydrochloride at pH 7.0 with a maximum concentration of 0.5 mg/mL and filtered by 0.22 μm Millipore membranes. Then a three-step dialysis process was performed at 4 °C to fully refold the target protein. Each step takes 3 h. The buffer used in each step is displayed below. Step 1: 50 mM $Na_2HPO_4$, 50 mM NaCl, 2 M Guanidine hydrochloride, pH 7.0 with 5% glycerin (2 L); Step 2: 50 mM $Na_2HPO_4$, 50 mM NaCl, 1 M Guanidine hydrochloride, pH 7.0 with 2.5% glycerin (4 L); step 3: 50 mM $Na_2HPO_4$, 50 mM NaCl, pH 7.0 (5 L). Then the dialysate was filtered and concentrated before further purification with a Superdex 75 gel filtration column (GE Healthcare) in a buffer of 25 mM $Na_2HPO_4$, 50 mM NaCl, pH 7.0. Fractions containing the target protein were concentrated and stored at −80 °C. The purity was

assessed by SDS-PAGE. Protein concentration was determined by BCA assay (Thermo Fisher).

## NMR spectroscopy

All NMR experiments were collected at 298 K on a Bruker Avance 900 MHz spectrometer equipped with a cryogenic TXI probe. All protein samples were prepared using the same NMR buffer of 25 mM $Na_2HPO_4$, 50 mM NaCl, and 10% (v/v) $D_2O$ at pH 7.0. For the titration samples, $^{15}$N-labeled A1E1 was mixed with L3D2/L3D3 at the molar ratio of 1–2, and each sample was made to a total volume of 500 μL containing 35 μM $^{15}$N-A1E1 in the absence and presence of unlabeled L3D2/L3D3 that diluted from high concentration stocks. Bruker standard pulse sequence (hsqcetfpf3gpsi) was used to collect the 2D $^1$H-$^{15}$N HSQC spectra, with a spectra width of 16 and 30 ppm for proton and nitrogen dimensions. Chemical shift deviations (CSD) were calculated using the Eq. 1 below,

$$CSD = \sqrt{(\Delta\delta1H)^2 + 0.0289(\Delta\delta15N)^2} \tag{1}$$

Where $\Delta\delta1H$ and $\Delta\delta15N$ are the chemical shift differences of amide proton and amide nitrogen between the free and bound state of A1E1, respectively. All NMR spectra were processed using NMRPipe[72] (https://www.ibbr.umd.edu/nmrpipe/index.html) and analyzed using NMRView[73] (http://docs.nmrfx.org/viewer/basics/guide).

## Live-cell imaging

α-Syn PFF were labeled with pHrodo dye (Invitrogen, USA), exhibiting minimal fluorescence at neutral pH, and enhanced fluorescence in low pH. WT and Aplp1$^{-/-}$ neurons 9–11 days in vitro were infected with lentiviral particles of Aplp1 expression or control, 3 days prior to the addition of α-syn-pHrodo PFF. Live-cell images were recorded by Microscope Axio Observer Z1 (Zeiss, USA) as previously described[20]. The baseline was established as the fluorescence intensity of the neuron at 2–3 min after α-syn-pHrodo PFF administration. The internalized α-syn-pHrodo PFF at each time point was obtained by tracking individual neuronal objects, images were collected at 30 sec intervals. The signal of internalized α-syn-pHrodo PFF was achieved by subtracting the intensity of the baseline in each experiment. All the data acquisition and analysis were performed with the same setting.

## Immunocytochemistry

Immunocytochemistry for the expression of Aplp1 and Lag3 was performed on floating sections using antibodies listed. The techniques were previously described[74]. Briefly, sections were incubated overnight at 4 °C in primary antibodies diluted in 0.1 M PBS (pH 7.4), containing 0.1% Triton and 10% normal goat serum. After washing with PBS, sections were incubated with secondary antibodies for 1 h at room temperature and then mounted. Z-stacks of 0.75 μm-thick optical sections through the entire thickness of the slice were captured using a confocal microscopy Zeiss 880 provided by MPI CORE, NINDS, JHU Center for Neuroscience Research under 40x oil objective. Measurements were taken from 3 pairs of mice ($n = 3$) in each group and handled with ImageJ software. The results are presented as a mean ± SEM, and the Student's unpaired *t*-test was performed to test the hypothesis of means' equality.

## Co-localization of endosome markers and α-syn-biotin PFF

Primary neurons were transfected with FLAG-Aplp1 expression vector 2 days prior to the addition of α-syn-biotin PFF (final concentration 1 μM). After 2-h of incubation, the neurons were fixed and immunostained, and the images were obtained using the same exposure time and treated in the same way for analysis. The signal of α-syn-biotin PFF co-localized with endosome markers (e.g., Rab5, Rab7) was measured and quantified by the Zeiss Zen software (https://www.zeiss.com/

microscopy/en/products/software/zeiss-zen-lite.html; *RRID:SCR_023747*) as previously described[20].

### Endolysosome enrichment

A-Syn-biotin PFF was administered to neuron (12–14 days in vitro) cultures and incubated for 1.5 h (final concentration 1 μM). Neurons were washed with a pre-warmed culture medium followed by 30 s trypsin to remove the bound α-syn-biotin PFF. Endolysosomes were enriched as previously described[20]: the neurons were harvested with PBS and prepared with lysis buffer (250 mM sucrose, 50 mM Tris-HCl [pH 7.4], 5 mM $MgCl_2$, 1 mM EDTA, 1 mM EGTA) with a protease inhibitor cocktail (Roche, USA). The suspended cell lysates were pipetted 6 times gently and passed through a syringe 20 times (1 ml TB Syringe, BD, USA), and avoid any bubbles in the procedures. The endolysosomes were harvested in the third pellet followed by three steps of centrifugation 1st ($1000 \times g$, 10 min), 2nd ($16,000 \times g$, 20 min) 3rd ($100,000 \times g$, 1 h) for immunoblot analysis.

### Uptake of α-syn-biotin PFF in SH-SY5Y cells with Aplp1-Lag3 overexpression

SH-SY5Y were transfected with pcDNA3.1-h*APLP1* and pcDNA3.1-h*APLP1*(mut9) individually, with/without Lag3 transfection. Two days after transfection, SH cells were incubated with α-Syn PFF biotin (31 nM, 500 nM, 1 μM) for 2 hr at 37 °C. The cells were treated with 0.25% trypsin for 10 s after removing the culture medium. The cells were washed three times with PBS, and then fixed with 4% PFA for 15 min. The cells were permeabilized in 0.2% Triton X-100 for 10 min at room temperature following the incubation with anti-FLAG 1:1000 (Cell Signaling Technology, Cat# 14793s) and anti-Myc 1:1000 (Cell Signaling Technology, Cat# 2276s) for 16 h at 4 °C, then washed and incubated with the secondary antibody.

### Tissue lysate preparation and Western blot analysis

Dissected brain regions of interest or culture samples were homogenized with TX-soluble buffer (50 mM Tris [pH 8.0], 150 mM NaCl, 1% Triton X-100) containing protease and phosphatase inhibitors (Roche, USA). The supernatants were collected for soluble fraction after centrifugation ($20,000 \times g$, 20 min), and the pellets were resuspended in TX-insoluble buffer (containing 2% SDS) with protease and phosphatase inhibitors. Protein concentrations of samples were determined using the BCA assay (Pierce, USA), and samples (10–20 μg total proteins) were loaded on SDS-polyacrylamide gels (12.5 13.5%) and transferred onto nitrocellulose membranes. Blots were blocked in 5% nonfat milk or 5% BSA in TBS-T (Tris-buffered saline, 0.1% Tween 20) and probed using various primary antibodies and related secondary antibodies, and then were detected using ECL or SuperSignal Femto substrate (Thermo Fisher, USA) and imaged by ImageQuant LAS 4000mini scanner (GE Healthcare Life Sciences, USA) or by film. A list of antibodies used is provided in Supplementary Table 2.

### Co-immunoprecipitation (co-IP)

For the interaction of Aplp1 and Lag3, brain lysates were prepared, or HEK293FT (RRID:CVCL_6911) cells were transfected with receptors of interest or deletion mutants. The cell or brain samples were homogenized with lysis buffer containing 50 mM Tris [pH 8.0], 150 mM NaCl, 1% Triton X-100, and protease inhibitors (Roche, USA). After the centrifugation ($20,627 \times g$, 20 min), protein concentration of the supernatants was determined using the BCA assay (Pierce, USA). Aliquots of the samples containing 500 μg of protein were incubated with the appropriated microbeads, such as Dynabeads MyOne Streptavidin T1 (Invitrogen, USA), Dynabeads Protein G (Life Technologies, USA) incubated with Myc (Cell Signaling Technology, Cat# 2276s), 410C9 or CT11 antibody, and FLAG M2 Magnetic Beads, (Sigma, USA). The samples were pre-cleared with the appropriated beads for one hour at room temperature (RT), and meanwhile, the related beads were

incubated for one hour (10 min for Myc) with the appropriated antibodies and control antibodies. Pre-cleared samples were incubated with microbeads overnight at 4 °C or 1 h at RT (for Myc). The IP complexes were washed 4–6 times with IP buffer and then denatured by adding 2× Laemmli Buffer plus β-mercaptoethanol.

### Microfluidic chambers

Triple compartment microfluidic devices (TCND1000, Xona Microfluidic, LLC, USA) were attached on glass coverslips and coated with 2.5 mg/ml poly-L-lysine overnight and washed with 0.1% Triton X-100 and dd-water three times each, and then replaced with Neurobasal media. Approximately 10,000 neurons were plated per chamber. At 7 days in vitro, 0.5 μg α-syn PFF were added in chamber 1 of all the four groups. To prevent α-syn PFF diffusion due to the flow direction, a 50 μl difference in media volume was maintained between chamber 1 and chamber 2, and chamber 2 and chamber 3 according to the manufacturers' instructions. Neurons were fixed 14 days after administration of α-syn PFF using 4% PFA in PBS, followed by immunofluorescence staining.

### Cell death assessment

Primary cortical neurons were treated with 5 μg/mL of α-syn PFF for 14–21 days. Percent of cell death was determined by staining with 7 μM Hoechst 33342 and 2 μM propidium iodide (PI) (Invitrogen, USA). PI was diluted in a warm neuronal culture medium and incubated for 5 min and then images were taken by a Zeiss microscope equipped with automated computer-assisted software (Axiovision 4.6, Carl Zeiss, Dublin, CA (https://www.micro-shop.zeiss.com/en/us/system/software-axiovision+software-products/1007/; RRID:SCR_002677)). The background signals were subtracted with the intensity of the control group, and the percent of neuronal death was calculated by PI signal divided by the Hoechst 33342 signal. See https://doi.org/10.17504/protocols.io.n92ldmd9ol5b/v1.

### Stereotaxic injection and sample preparation

On the day of intrastriatal injections, α-syn PFF was diluted in sterile PBS and briefly sonicated. Two-three months-old mice were anesthetized with a mixture of ketamine (100 mg/kg) and xylazine (10 mg/kg), followed with intrastriatal injection with α-syn PFF (5 μg/2 μL at 0.4 μL/min) at the coordinates of the dorsal striatum: anteroposterior (AP) = +0.2 mm, mediolateral (ML) = +2.0 mm, dorsoventral (DV) = +2.8 mm from bregma. Injections were performed using a 2 μL syringe (Hamilton, USA), and the needle was maintained for an additional 5 min for complete absorption before slow withdrawal of the needle. After surgery, animals were monitored and post-surgical care was provided. Behavioral tests were performed 180 days after α-syn PFF injection, and then mice were euthanized for biochemical and histological studies. For biochemical studies, tissues were immediately dissected and frozen at −80 °C. For histological studies, mice were perfused with PBS and 4% PFA, and brains were removed, followed by fixation in 4% PFA overnight and transfer to 30% sucrose for cryoprotection. See https://doi.org/10.17504/protocols.io.dm6gp34p1vzp/v1.

### Lag3 mRNA detection by RNAScope

RNAscope in situ hybridization was performed according to the manufacturer's instructions (RNAScope multiplex fluorescent assay, Cat No. 323110, Advanced Cell Diagnostics). Briefly, fresh-frozen brains of 3–4 months-old mice were sectioned at 12 μm, post-fixated in PFA, and dehydrated with ETOH according to the manufacturer's instructions. These steps were followed by Hydrogen Peroxide (10 min) and Protease IV (30 min) treatments. Sections were incubated for 2 h in a HybEZ oven (Advanced Cell Diagnostics) at 40 °C with probes targeting Lag3 (Cat No. 1221971), Tmem119 (Cat No. 472901) and Map2 (Cat No. 431151) mRNA. Signals from bound

probes were amplified and then labeled by fluorophores (Multiplex assay V2, Cat No. 323110, Advanced Cell Diagnostics). For segmentation and automated counting, the Cell Profiler program was used. RNAscope amplifies signals and detects them as dots and a Cell profiler is used to detect those dots to classify cells accordingly. Dots within the 7 μm (20 pixels) distance from each nucleus are registered to it. To identify a cell as a neuron, the minimum requirement for the number of MAP2 dots within the mentioned distance to its nucleus is 7. The same requirement for TMEM119 dots to define a cell as microglia is 3. The presence of a Lag3 dot in microglia registers them as Lag3+ microglia. The same rule applies to neurons but the neurons that have even one TMEM119 dot in close proximity to their nuclei (7 μm) are rejected and NOT classified as Lag3 positive neurons but as Lag3 negative neurons.

### Behavioral tests

To evaluate α-syn PFF-induced behavioral deficits, PBS- and α-syn PFF-injected mice were assessed by pole test and cylinder test. The experimenter was blinded to the treatment group for all behavioral studies. All tests were recorded and performed between 10:00 and 16:00 in the lights-on cycle.

### Pole test

Mice were acclimatized in the behavioral procedure room for 30 min. The pole was made of a 75-cm metal rod (diameter 9 mm) and wrapped with bandage gauze. Mice were placed near the top of the pole (7.5 cm from the top of the pole) facing upwards. The total time taken to reach the base of the pole was recorded. Before the actual test, mice were trained for three consecutive days. Each training session consisted of three test trials. On the test day, mice were evaluated in three sessions and the total time was recorded. The maximum cutoff time to stop the test and recording was 120 s. Results for turn down, climb down and total time were recorded. See https://doi.org/10.17504/protocols.io.5jyl8938rv2w/v1.

### Cylinder test

Spontaneous movement was measured by placing animals in a small transparent cylinder (height, 15.5 cm; diameter, 12.7 cm). Spontaneous activity was recorded for 10 min. The number of forepaw touches, rears, and grooming were measured. Recorded files were viewed and rated in slow motion by an experimenter blinded to the mouse type. See https://doi.org/10.17504/protocols.io.5qpvo3jmzv4o/v1.

### Grip strength

Neuromuscular strength was measured by the maximum holding force generated by the mice (Bioseb, USA). Mice were allowed to grasp a metal grid either by their fore and/or hind limbs or both. The tail was gently pulled and the maximum holding force was recorded by the force transducer when the mice released their grasp on the grid. The peak holding strength was digitally recorded and displayed as a force in grams. See https://doi.org/10.17504/protocols.io.3byl4qmoovo5/v1.

### Purification of anti-Lag3 410C9 and labeling

The 410C9 hybridoma cell line is available from Dr. Dario Vignali under a material transfer agreement with the University of Pittsburgh. We cultured the 410C9 hybridoma cell line and collected the supernatant culture medium. The supernatant was concentrated and purified over the HiTrap Protein G HP antibody purification column (GE Healthcare). To visualize 410C9 in vivo, a near infrared (IR) dye IRDye680RD was used to label 410C9. The 410C9 solution was mixed with IRDye680RD NHS Ester (molar ratio 1:2). The mixture solution was consistently shaken on the shaker at 4 °C for 16 h, and then dialyzed against DI water for 24 h using the dialysis membrane with molecular weight cutoff of 3500 Da to remove any free dye. The obtained solution was subjected to BCA assay to determine the concentration of 410C9 for further study.

### Statistical analysis and reproducibility

All data were analyzed using GraphPad Prism version 8.0 for Mac (GraphPad Software, San Diego, CA, USA, http://www.graphpad.com/; RRID:SCR_002798). Statistics Data are presented as the mean ± SEM. With at least 3 independent experiments. Representative morphological images were obtained from at least 3 experiments with similar results. Statistical significance was assessed via a one- or two-way ANOVA test followed by indicated post-hoc multiple comparison analysis. Assessments with $p < 0.05$ were considered significant.

The sample size for both cell and animal experiments was determined to be adequate based on the experience of previous studies and literature describing similar experiments, as well as the consistency of measurable differences between experimental groups. The number of mice analyzed is reported in the figure legends or supplementary methods of the respective sections. For in vitro experiments, experiments were conducted in triplicate and all replicates obtained similar results. Samples or animals were excluded from analysis in cases of technical failure. Random assignment of cells and animals to different experimental groups was implemented to minimize bias. Moreover, the animals used in the experiments were littermates. Mice were grouped according to genotype and randomly allocated for different experiments. To minimize bias, investigators were blinded during the allocation process, the conduct of the experiment, outcome assessment, and data analysis.

### Reporting summary

Further information on research design is available in the Nature Portfolio Reporting Summary linked to this article.

## Data availability

Further information and requests for resources and reagents should be directed to and will be fulfilled by the Lead Contact, Ted M. Dawson (tdawson@jhmi.edu). There are no restrictions on any data or materials presented in this paper. All data are available in the main text or the Supplementary file. Source files are available at Dryad: https://doi.org/10.5061/dryad.5hqbzkhcw. Source data are provided with this paper.

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

## Acknowledgements

We thank I.-H. Wu and Hao Wen (Gilbert) Chen for graphic art assistance. We appreciate the Aplp1 antibodies as a gift provided by Dr. Gopal Thinakaran at the University of South Florida Morsani College of Medicine and the support from Dr. Haiquan Mao at the Johns Hopkins University. The authors acknowledge the joint participation by the Adrienne Helis Malvin Medical Research Foundation through its direct engagement in the continuous active conduct of medical research in conjugation with the Johns Hopkins Hospital and the Johns Hopkins University School of Medicine and the Foundation's Parkinson's Disease Program M-2014. T.M.D. is the Leonard and Madlyn Abramson Professor in Neurodegenerative Diseases. The Multiphoton Imaging Core of Johns Hopkins University was used (NS050274) in some of the imaging studies. This work was supported by the funding from NIH R01NS107318, R01AG073291, R01AG071820, RF1NS125592, K01AG056841, R21NS125559, Parkinson's Foundation PF-JFA-1933, Maryland Stem Cell Research Foundation 2019-MSCRFD-4292, American Parkinson's Disease Association (X.B.M.); NIH R01 NS107404 (H.S.K.), NIH P01 AI108545, R01 AI144422 (D.A.A.V., C.J.W.); Uehara Memorial Foundation, Japan (Y.K.); JPB Foundation (T.M.D.), Adrienne Helis Malvin Medical Research Foundation, M-2014 (T.M.D., V.L.D.); Parkinson's Disease Foundation, PDF-SFW-1572 (P.G.). American Parkinson Disease Foundation, PDF-APDA-SFW-1650 (P.G.).

## Author contributions

X.B.M., H.G., and D.H.K. led the project and contributed to all aspects of the study. Y.K., N.W., E.Q.X., R.K., X.T.M., H.B.W., C.C., Y.Q.L., H.T.B., F.A., Q.C., L.G.J., H.H.H., S.H.L., M.C., A.L., J.Y., C.R., M.J.S., P.G., S.B., S.J.K., S.Z., H.Q.L., S.K., and M.Y.Y. contributed to biochemical, cellular, mouse experiments and revision work. S.S.K. contributed to HPLC analysis. X.K., Y.S., M.S., C.J.W., D.A.A.V., and U.C.M. provided key reagents. S.N.Z., C.Y.J., and C.L. contributed to NMR studies. X.B.M., H.S.K., V.L.D., and T.M.D. designed research, X.B.M., H.G., D.H.K., Y.K., N.W., E.Q.X., and R.K. analyzed data, and X.B.M., H.S.K., V.L.D., and T.M.D. wrote and revised the paper. All authors reviewed, edited, and approved the paper. Y.K., N.W., E.Q.X., and R.K. contributed equally to this work.

## Competing interests

D.A.A.V. and C.J.W. have submitted patents on Lag3 that are approved or pending and are entitled to a share in net income generated from licensing of these patent rights for commercial development. Patents relate to the LAG3 aspects of this manuscript (Fig. 4, Supplementary Figs. 5, 6, and 9 are as follows: US Provisional Patent Application #60/451,039 (St Jude # SJ-02-0027)). Title: Regulating T-Cell Homeostasis. Priority date: February 28, 2003. Pardoll D, Vignali DAA, Powell J, Drake C, Huang C-T, Workman CJ. US Provisional Patent Application #60/482,143 (St Jude # SJ-03-0012 / JH # Dm-4255). Title: Modulating Regulatory T-Cell Activity via CD223. Filed: June 24, 2003. Combined and published September 16, 2004, as International Application No. WO 2004/078928 entitled 'T-Cell Regulation'. European Patent Application No: 07021595.9; publication no: 1897548, Feb 13, 2008. Pending U.S. application for "T-cell Regulation" published Oct. 26, 2006, as Pub. No. 2006/0240024 and corresponding foreign applications pending in Canada and Japan. Patents granted in US (8551481 - issued 10/8/2013; 9005629 - issued 04/14/2015; 10787513 - issued 09/29/2020; 10934354 - issued 03/02/2021), Australia (2004217526; issued – 8/12/2010), Europe (1897548; issued – 08/14/2013), Japan (6758259; issued 09/03/2020), and Hong Kong (1114339; issued 11/22/2013). Additional applications pending in Canada (1), Europe (2), Japan (2), and the U.S. (1). The remaining authors declare no competing interests.

## Additional information

[1]Neuroregeneration and Stem Cell Programs, Institute for Cell Engineering, Johns Hopkins University School of Medicine, Baltimore, MD 21205, USA. [2]Department of Neurology, Johns Hopkins University School of Medicine, Baltimore, MD 21205, USA. [3]Adrienne Helis Malvin Medical Research Foundation, New Orleans, LA 70130-2685, USA. [4]Interdisciplinary Research Center on Biology and Chemistry, Shanghai Institute of Organic Chemistry, Chinese Academy of Sciences, 26 Qiuyue Road, Shanghai 201210, China. [5]University of the Chinese Academy of Sciences, 19 A Yuquan Road, Shijingshan District, Beijing 100049, China. [6]Institute for NanoBioTechnology, Johns Hopkins University, Baltimore, MD 21218, USA. [7]Department of Materials Science and Engineering, Whiting School of Engineering, Johns Hopkins University, Baltimore, MD 21218, USA. [8]Department of Bioengineering, Nagaoka University of Technology, 1603-1 Kamitomiokamachi, Nagaoka, Niigata 940-2188, Japan. [9]Institute for Pharmacy and Molecular Biotechnology IPMB, Department of Functional Genomics, University of Heidelberg, Im Neuenheimer Feld 364, 69120 Heidelberg, Germany. [10]Hugo W. Moser Research Institute at Kennedy Krieger, 707 North Broadway, Baltimore, MD 21205, USA. [11]Department of Immunology, University of Pittsburgh School of Medicine, Pittsburgh, PA 15261, USA. [12]Tumor Microenvironment Center, UPMC Hillman Cancer Center, Pittsburgh, PA 15232, USA. [13]Department of Physiology, Johns Hopkins University School of Medicine, Baltimore, MD 21205, USA. [14]Solomon H. Snyder Department of Neuroscience, Johns Hopkins University School of Medicine, Baltimore, MD 21205, USA. [15]Department of Pharmacology and Molecular Sciences, Johns Hopkins University School of Medicine, Baltimore, MD 21205, USA. [16]Present address: Department of Neurology, Nanjing Brain Hospital, Nanjing, Jiangsu 210029, PR China. [17]Present address: Medical College, Yangzhou University, Yangzhou, Jiangsu 225001, PR China. [18]Present address: Department of Anesthesiology, West China Hospital, Sichuan University. The Research Units of West China (2018RU012)-Chinese Academy of Medical Sciences, West China Hospital, Sichuan University, Chengdu, Sichuan 610041, PR China. [19]Present address: Department of Brain and Cognitive Sciences, Massachusetts Institute of Technology, Cambridge, MA 02139, USA. [20]Present address: Department of Pharmacology, College of Medicine, Dong-A University, 32 Daesin Gongwwon-ro, Seo-gu, Busan 49201, Republic of Korea. [21]Present address: Department of Biological Science and Biotechnology, Chungbuk National University, Cheongju, Chungbuk 28644, Republic of Korea. [22]Present address: Department of Physiology, School of Basic Medical Sciences (Institute of Basic Medical Sciences), Shandong First Medical University & Shandong Academy of Medical Sciences, Jinan 250000, China. [23]Present address: Picower Institute for Learning and Memory, Cambridge, MA 02139, USA. [24]Present address: Harvard-MIT MD/PhD Program, Harvard Medical School, Boston, MA 02115, USA. [25]These authors contributed equally: Xiaobo Mao, Hao Gu, Donghoon Kim.
✉e-mail: xmao4@jhmi.edu; hko3@jhmi.edu; vdawson1@jhmi.edu; tdawson@jhmi.edu

