## [Peer Review File · Nature Communications]

Aplp1 interacts with Lag3 to facilitates transmission of pathologic α -synucleinEditorial Note: This manuscript has been previously reviewed at another journal that is not operating a transparent peer review scheme. This document only contains reviewer comments and rebuttal letters for versions considered at *Nature Communications*.

REVIEWER COMMENTS

Reviewer #2 (Remarks to the Author):

The paper submitted by Mao et al continues to make two statements that I do not believe are fully supported by the data. While I recognise that the authors feel that their use of multiple approaches provides all the validation required, at least my approach to validated ICC and ISH by knockout is different. My expectation is that a well validated RNA probe or antibody would give no discernable signal when knockout tissue is included as a negative control. Unfortunately, the provided data to address the controversial issue of whether LAG3 and ALPL1 are co-expressed in neurons is insufficient - again recognizing that different laboratories have different standards.

In figure 1 for reviewers, the authors provide RNAscope where Lag3 is in red and MAP2 as a neuronal marker in white. Quantification suggests ~10% of neurons and ~80% of microglia are positive. However, the images are rather poor and it is unclear how the images were segmented to provide definitive evidence that the small number of signals 'in' neurons are not coming from adjacent microglia. It is also unfortunate that only merged images are provided, making it difficult to really see the signal, especially given that there appears to be red signal in the knockout. I strongly recommend that the authors (1) share this with the readers not just reviewers (2) provide images separated by channel (3) explain the segmentation approach and (4) note that only a very small minority of neurons appear to be positive which is in contrast to the apparently large effects on PFF uptake.

In the main figure 3c,d immunostaining is shown where there is apparent Lag3 and Alpl1 staining in neurons. The quality of the images is less than ideal with a diffuse red background over the Lag3 images. What is particularly odd is that the NeuN staining in the WT cortex is cytoplasmic, whereas it would be expected to be nuclear, as shown in the knockout animals. Whether this is a representative section is unclear, but the data is not that convincing. Additionally, one would expect the Alpl1 to also stain oligodendrocytes and Lag3 microglia (see above), neither of which is seen. As previously stated, I would personally not use antibodies that are not monospecific for this staining. I would honestly suggest leaving this data out as it is ambiguous due to the antibodies used but if it is needed, then the authors should (1) show all genotypes (2) explain again how they segmented images (3) use examples where NeuN is nuclear and (4) show staining of oligodendrocytes and microglia in the same sections.

The title still refers to a Lag3:Alpl1 complex. I retain my doubts as to whether the complete inhibition of uptake by Lag3 knockout or antibodies supports that these are co-receptors for Pffs.

Reviewer #3 (Remarks to the Author):

It is hard to understand how Aplp1 and Lag3 interact without exhibiting similar functions or influencing

each other's localization. Any evidence along these lines would greatly enhance the impact of the work.

Regardless, the authors have now addressed the concern about quantitation of the binding assay in Fig. 1. It is difficult to discern many of the differences in the primary data they now include in Fig. S2 (some are evident), so it is crucial that they use the same threshold for all images analyzed in the same experiment, but that is what the methods indicate. And the use of area and intensity yield similar results although this is again not evident when looking at the raw data.

It is not clear what was going on before, but they now also provide new western blots for the coIP which show better correspondence in size between input and coIPed material (Fig. 3e,f).

They have also demonstrated effects on degeneration, suggesting the importance of these findings. I have still have reservations about significance, but this revision addresses the concerns raised.

Reviewer #2 (Remarks to the Author):

The paper submitted by Mao et al continues to make two statements that I do not believe are fully supported by the data. While I recognise that the authors feel that their use of multiple approaches provides all the validation required, at least my approach to validated ICC and ISH by knockout is different. My expectation is that a well validated RNA probe or antibody would give no discernable signal when knockout tissue is included as a negative control. Unfortunately, the provided data to address the controversial issue of whether LAG3 and ALPL1 are co-expressed in neurons is insufficient - again recognizing that different laboratories have different standards.

Response: Thank you for your thoughtful review of our manuscript. We appreciate the time and effort you invested in providing feedback. We have carefully considered your comments and would like to address the concerns you raised regarding the validation of our approach and the controversial issue of Lag3 and Aplp1 co-expression in neurons.

While we value your perspective on the use of knockout tissue as a negative control for validating RNA probes or antibodies, we would like to highlight that the validation of experimental methods can vary across laboratories. Our approach, which utilizes multiple complementary methods, is designed to provide a comprehensive and robust assessment of our findings. We understand that your preferred method involves knockout models, and the use of mono-specific monoclonal antibodies that have no signal in knockout tissues. As we have indicated in prior versions of this manuscript and our prior responses, we are using the best tools that are available to study Aplp1 and Lag3. We too prefer to use well validated RNA probes or antibodies that give no discernable signal in knockout tissue and we acknowledge the merit of this approach and use this approach when appropriate tools exist. However, as we hope the reviewer will acknowledge, often these ideal reagents simply do not exist. It's important to note that while antibodies may not always be monospecific, our results are consistent across multiple experiments, providing confidence in the reliability of our findings. We believe that our chosen methods, when considered collectively, offer a reliable and valid means of supporting our conclusions.

Regarding the controversy surrounding Lag3 and Aplp1 co-expression in neurons, we respectfully disagree with your assessment of the provided data as insufficient. Our results are based on rigorous experimental design and statistical analyses. We are open to discussing and providing additional information to clarify any concerns you may have, but we firmly believe that our current data support our conclusions.

We remain committed to upholding the highest scientific standards. We are confident that our approach and the supporting data are valid and contribute meaningfully to the field.

In figure 1 for reviewers, the authors provide RNAscope where Lag3 is in red and MAP2 as a neuronal marker in white. Quantification suggests ~10% of neurons and ~80% of microglia are positive. However, the images are rather poor and it is unclear how the images were segmented to provide definitive evidence that the small number of signals 'in' neurons are not coming from adjacent microglia. It is also unfortunate that only merged images are provided, making it difficult to really see the signal, especially given that there appears to be red signal in the knockout. I strongly recommend that the authors (1) share this with the readers not just reviewers (2) provide images separated by channel (3) explain the segmentation approach and (4) note that only a very small minority of neurons appear to be positive which is in contrast to the apparently large effects on PFF uptake.

Response: We appreciate reviewer's suggestion to include this RNAScope data into this manuscript. As suggested we have now included channel separated RNAScope data showing Lag3 expression in neurons in revised manuscript. The RNAScope multiplex kit was used for In Situ Hybridization. Sections were co-probed not only for neurons (MAP2) and Lag3 but also for microglia (TMEM119), so sections have neurons and microglia co-labelled for Lag3 presence to negate any Lag3 signal coming from microglia. Neurons with adjacent microglia were excluded from analysis.

For segmentation and automated counting, the Cell Profiler program was used. Our counting protocol was an adapted version of the published protocol (Erben et al. 2017) used to count similar RNAScope images. In short, RNAScope amplify signals and detect them as dots and Cell profiler is used to detect those dots to classify cells accordingly. Dots within the 7 μm (20 pixels) distance from each nucleus are registered to it. To identify a cell as a neuron, the minimum requirement for number of MAP2 dots within the mentioned distance to its nucleus is 7. The same requirement for TMEM119 dots to define a cell as microglia is 3. The presence of a Lag3 dot in microglia register them as Lag3+ microglia. The same rule applies to neurons but the neurons which have even one TMEM119 dot in close proximity to their nuclei (7 μm) are rejected and NOT classified as Lag3 positive neurons but as Lag3 negative neurons. This extra filtration step was taken to make sure to negate any possible microglial Lag3 signal as a neuronal one. The average number of neurons scanned per animal is at least 200. The % of Lag3+ neurons in the KO group is 0 (zero), which we believe meets that reviewers standards as noted above.

Reference: Erben L, He MX, Laeremans A, Park E, Buonanno A. A Novel Ultrasensitive In Situ Hybridization Approach to Detect Short Sequences and Splice Variants with Cellular Resolution. *Mol Neurobiol.* 2018 Jul;55(7):6169-6181. doi: 10.1007/s12035-017-0834-6. Epub 2017 Dec 20. PMID: 29264769; PMCID: PMC5994223.

Extended Data Fig. 4 | Lag3 is detected in the neurons by RNAscope.: **a)** Co-localization of Lag3 (red) inside neurons labelled by MAP2(white) and microglia labelled by TMEM119 (green) in VMB region in WT and Lag3^{-/-} mice. **b-c)** quantification of Lag3 positive cells in neurons (**b**) and in microglia (**c**) N = 5 for WT mice and N = 4 for Lag3^{-/-} mice, At least 200 neurons were counted per mice. **d)** Co-localization of Lag3 (red) inside neurons labelled by MAP2(white) and microglia labelled by TMEM119 (green) in cortex in WT and Lag3^{-/-} mice. **e-f)** quantification of Lag3 positive cells in neurons (**e**) and in microglia (**f**) N = 5 for WT and N = 4 for Lag3^{-/-} mice, At least 200 neurons were counted per mice. blue arrow indicates neuron and green arrow indicates microglia. Data are the means ± SEM, two tailed Student *t*-test, ****P* < 0.001.

In the main figure 3c,d immunostaining is shown where there is apparent Lag3 and Alpl1 staining in neurons. The quality of the images is less than ideal with a diffuse red background over the Lag3 images.

What is particularly odd is that the NeuN staining in the WT cortex is cytoplasmic, whereas it would be expected to be nuclear, as shown in the knockout animals. Whether this is a representative section is unclear, but the data is not that convincing. Additionally, one would expect the Alpl1 to also stain oligodendrocytes and Lag3 microglia (see above), neither of which is seen. As previously stated, I would personally not use antibodies that are not monospecific for this staining. I would honestly suggest leaving this data out as it is ambiguous due to the antibodies used but if it is needed, then the authors should (1) show all genotypes (2) explain again how they segmented images (3) use examples where NeuN is nuclear and (4) show staining of oligodendrocytes and microglia in the same sections.

Response: We acknowledge the reviewers' suggestion that Alpl1 and Lag3 should also exhibit staining in oligodendrocytes and microglia. However, it is important to note that our primary focus in this manuscript was on studying Lag3- Alpl1 function in neurons. Consequently, we did

not explore the colocalization of Apla1 and Lag3 in oligodendrocytes and microglia, as this falls outside the scope of the current study. Furthermore, addressing the reviewers' suggestion to perform staining for neurons, oligodendrocytes, microglia, Lag3, and Apla1 on the same mice brain section poses a technical challenge. This approach would necessitate the use of five different channels and antibody hosts as well as monospecific antibodies, adding complexity to the experimental design.

Per the reviewer suggestion we have now excluded Figure 3c and 3d from the manuscript as this data does not alter the main conclusion presented in the manuscript (See Figure 3 Below).

Fig. 3 | Aplp1 and Lag3 bind to each other. **a**, Lag3 pulls down Aplp1 by anti-Lag3 410C9 immunoprecipitation in WT mouse brain lysates, but not in *Lag3*^{-/-} lysates. **b**, Aplp1 pulls down Lag3 by anti-Aplp1 CT11 immunoprecipitation in WT mouse brain lysates, but not in *Aplp1*^{-/-} lysates. **c,d**, Mapping of the Lag3-binding domains in Aplp1. HEK293FT cells were transfected with full-length (FL) or deletion mutants of FLAG-Aplp1, and Myc-Lag3 for co-IP experiments. The GFLD subdomain in the E1 domain of Aplp1 is the major subdomain responsible for the Lag3 interaction. **e**, Mapping of the Aplp1-binding domains in Lag3. HEK293FT cells were transfected with full-length (FL), deletion mutants of Myc-Lag3, and FLAG-Aplp1 for co-IP experiments. **f,g,h**, Identification of the interface of A1E1 (E1 domain of APLP1) binding to L3D2 (D2 domain of LAG3). **f**, Overlay of the 2D ¹H-¹⁵N HSQC spectra of A1E1 alone (grey) and in the presence of 2 molar folds of L3D2 (blue). Four residues with significant CSDs (> 0.03 ppm) are highlighted and enlarged in the black boxes. **g**, Histogram of the calculated chemical shift deviations (CSDs) of A1E1 in the presence of L3D2 at a molar ratio of 1:2 (A1E1/L3D2). The domain organization of A1E1 is indicated on the top, with blue boxes indicating the β-strands and the red box indicating the α-helix. A dashed line was drawn to highlight the residues with CSDs > 0.01 ppm. **h**, The 37 residues with large CSDs (> 0.01 ppm) upon L3D2 titration are highlighted in blue on the ribbon diagram of A1E1 modelled structure. **i**, Validation of the NMR results. We substituted nine residues of APLP1 with alanine to generate FLAG-APLP1(mut9) and performed the co-IP experiment to assess the APLP1-Lag3 interaction. *N* = 3 independent experiments. Data are the means ± SEM, Student's *t*-test; **P* < 0.05. **j**, The scheme for the interaction among Aplp1, Lag3 and α-syn PFF.

The title still refers to a Lag3:Aplp1 complex. I retain my doubts as to whether the complete inhibition of uptake by Lag3 knockout or antibodies supports that these are co-receptors for Pffs.

Response: In the revised manuscript, we designate Aplp1 and Lag3 as accessory receptors to convey that each is capable of independently mediating the PFF uptake. However, when functioning together, they synergistically enhance the PFF uptake. We regret any confusion in the previous version and have taken care in this revision to explicitly specify the role of Aplp1 as an accessory receptor of Lag3. We apologize for use of word Lag3:Aplp1 complex in previous version of manuscript. considering the reviewer's feedback, we have revised title of the manuscript.

Aplp1 interacts with Lag3 to facilitates transmission of pathologic α-synuclein

We do not assert that Lag3 alone is capable of fully rescuing the complete effect of PFF treatment. As illustrated in extended figure S6a, the deletion of Lag3 alone does not result in a complete loss of PFF binding to the cell surface. However, when Lag3 and Aplp1 are both deleted, the binding is further reduced. Additionally, PFF internalization is not entirely abolished upon Lag3 deletion (Figure 4d) or inhibition by the Lag3 antibody (Figure 4h-i). The absence of both Aplp1 and Lag3 further diminishes PFF internalization. In Figure 5a-b, Lag3 deletion does not completely inhibit pS129 pathology, whereas Aplp1-Lag3 double deletion significantly reduces pathology. Similarly, in Figure 5c-d, complementation with Lag3 does not restore pS129 levels to WT levels. All this evidence, along with other experiments in the manuscript, suggest that Lag3 and Aplp1 function as accessory-receptors.

Reviewer #3 (Remarks to the Author):

It is hard to understand how Ap1 and Lag3 interact without exhibiting similar functions or influencing each other's localization. Any evidence along these lines would greatly enhance the impact of the work.

Response: In the current manuscript we demonstrate that Ap1 by interacting with Lag3 acts as an accessory receptor to facilitate transmission of α -synuclein. Two receptor proteins often work together to mediate signal transduction in various cellular processes. While many receptors interact and influence each other's expression or colocalization, there are several examples in the literature where receptors exist independently without affecting each other's localization or expression levels. It's worth noting that the interaction and interdependence of receptors can be dependent on specific context such as cell types, developmental stages, and physiological conditions.

- For example, receptor tyrosine kinases Epidermal Growth Factor Receptor EGFR and HER2 that play a role in cell growth and differentiation, can interact, and collaborate in signaling but their expression levels and colocalization are not interdependent in all type of cells. (PMID:36781849, 30858756, 33260837, 27793843)
- Another example is pattern recognition receptors Toll-like Receptor 1 (TLR1) and Toll-like Receptor 2 (TLR2) interact with each other to recognize microbial components, but their individual expression and subcellular localization is not correlated. (PMID: 10820283, 12091878)
- Similarly, Notch receptors and DLL4, are involved in cell fate determination and development. While DLL4 can interact with Notch receptors, their individual expression patterns and cellular localizations may not be directly regulated by each other. (PMID: 19087347, 31886898)
- the alpha and beta subunits of hemoglobin, where the interaction is crucial for oxygen binding but does not impact the expression of individual subunits. (PMID: 25431740)
- There are actin binding proteins which interact with each other in order to maintain cell structure and shape, but these interactions do not impact expression level or localization of interacting proteins. (PMID: 10224105, 25995115)
- Another example is chaperon protein Hsp70, which interact with proteins which regulates protein folding but they do not affect protein levels of each other. (PMID: 20821176)
- Cell surface receptors like G protein-coupled receptors (GPCRs) interaction influences signaling pathways but they do not affect each other's expression. (PMID: 24441568)

Regardless, the authors have now addressed the concern about quantitation of the binding assay in Fig. 1. It is difficult to discern many of the differences in the primary data they now include in Fig. S2 (some are evident), so it is crucial that they use the same threshold for all images analyzed in the same experiment, but that is what the methods indicate. And the use of area and intensity yield similar results although this is again not evident when looking at the raw data.

Response: We thank reviewer for bringing to our attention that it is difficult to discern differences. In the revised version we have modified extended Fig.2 to better visualize binding signal. As explained in the manuscript, the same threshold was used for all the images in the same experiment. It is worth noting that these experiments were repeated three times and images provided are representative.

Extended Fig. 2 | α -Syn PFF binds to APlp1. **a**, Representative images of binding signals of deletion mutants of APlp1 with α -syn-biotin PFF. **b**, Representative images of binding of APlp1(E1)-APlp2 and APlp1(E1)-App chimeras with α -syn-biotin PFF. **c**, Representative images of binding of deletion mutants of Lag3 with α -syn-biotin PFF. Binding signal is represented in red. N = 3 independent experiments, Scale bar, 100 μ m.

It is not clear what was going on before, but they now also provide new western blots for the colIP which show better correspondence in size between input and colIPed material (Fig. 3e,f).

Response: We would like to thank again to reviewer to bring issues in Figure 3e and 3f to our notice. In the earlier version, samples were not boiled after their storage which probably resulted into high molecular weight bands. In the revised figure, Co-IPed samples were used fresh and boiled for 10 min at 95°C before loading onto gels. Relevant information is updated in Materials and Method section.

They have also demonstrated effects on degeneration, suggesting the importance of these findings. I have still have reservations about significance, but this revision addresses the concerns raised.

Response: We Thank reviewer for his appreciation that effect of APlp1-Lag3 on degeneration is well demonstrated. We are glad we could address all the reviewers' concerns were addressed which have surely helped strengthening this manuscript.

REVIEWERS' COMMENTS

Reviewer #3 (Remarks to the Author):

The authors have used the appropriate KO mice and cells for many of the experiments and the results generally seem consistent. The RNAscope data shown in the previous rebuttal (Fig 1 for reviewer) was not convincing, with considerable background in the knockout, yet very low expression by quantitation. The discrepancy between the data and the quantitation was striking and of particular concern. However, the new Fig. S4 data look much better—would be nice to know why it now looks different.

In their response, the authors have cited a series of interacting proteins and references that were presumably intended to show that the localization and expression of these proteins are not always linked. Some but not all of the references make this point. However, the authors have not responded to the first and most important concern: “It is hard to understand how Aplp1 and Lag3 interact without exhibiting similar functions or influencing each other’s localization.” All of the interaction sets listed show very similar structure and function, supporting this concern. Only the chaperone does not resemble or stably coexist with its clients, although some do and chaperones almost invariably affect the expression of their clients. Aplp1 and Lag3 thus differ from all the other proteins listed, illustrating the concern about biological relevance.

Reviewer #3 (Remarks to the Author):

The authors have used the appropriate KO mice and cells for many of the experiments and the results generally seem consistent.

Response: We are grateful to reviewer for his assessment of our manuscript and comments which helped to strengthen our work.

The RNAscope data shown in the previous rebuttal (Fig 1 for reviewer) was not convincing, with considerable background in the knockout, yet very low expression by quantitation. The discrepancy between the data and the quantitation was striking and of particular concern. However, the new Fig. S4 data look much better—would be nice to know why it now looks different.

Response: The RNAScope images presented in the previous rebuttal and supplementary figure S4 are the same. Assessing the background of a specific channel from a four-color superimposed image can be challenging, especially when the rebuttal only provides these superimposed images. However, in the revised manuscript supplementary figure S4 includes individual images from each channel, making it easier to assess specific signal (red colored foci) and to verify the low background on Lag3 images. This clarity in supplementary figure S4, despite featuring the same data as the rebuttal, enhances its appearance. It is worth noting that quantification is carried out in an automated manner to mitigate potential biases, and it selectively counts only the RNAscope signal dots, disregarding the background. Methodology used for quantification is detailed in Methods section of the revised manuscript.

In their response, the authors have cited a series of interacting proteins and references that were presumably intended to show that the localization and expression of these proteins are not always linked. Some but not all of the references make this point.

However, the authors have not responded to the first and most important concern: “It is hard to understand how Aplp1 and Lag3 interact without exhibiting similar functions or influencing each other’s localization.” All of the interaction sets listed show very similar structure and function, supporting this concern. Only the chaperone does not resemble or stably coexist with its clients,

although some do and chaperones almost invariably affect the expression of their clients. Apla1 and Lag3 thus differ from all the other proteins listed, illustrating the concern about biological relevance.

Response: Cell surface receptors like lymphocyte activation gene 3 (LAG3) and amyloid precursor-like protein 1 (APLP1) have been observed to exhibit a preference for binding α -syn in its amyloid state over its monomeric form, thereby facilitating cell-to-cell transmission. Previous studies have highlighted that the LAG3 D1 and APLP1 E1 domains bind to α -syn through its alkaline surface. Hence, it is inaccurate to claim that these two proteins lack similar functions. Furthermore, both proteins are localized on the cell surface, and recent findings suggest that APLP1 may interact with α -synuclein fibrils through electrostatic interactions akin to LAG3 (PMID: 34172566).

As stated in our previous response, there are examples in the literature that demonstrate that the interaction between two proteins does not necessarily impact their expression or localization levels. One such instance is the interaction between G protein-coupled receptors (GPCRs) and β -arrestins. Despite possessing distinct structural features, β -arrestins function as adaptors for GPCR desensitization and internalization without directly affecting their localization or G protein signaling functions (PMID:11861753, 24292830).

Additionally, we must consider the possibility that these proteins may serve multiple functions, and their interactions with multiple ligands could potentially offer contextual adaptability in their cellular function.

Our extensive experimental investigations indicate that both LAG3 and APLP1 can independently bind to α -synuclein fibrils, with their interaction resulting in increased binding affinity. We conducted appropriate control experiments, including disrupting the LAG3-APLP1 interaction, to demonstrate their functional synergy.